# Diffusion vs. direct transport in the precision of morphogen readout

Sean Fancher[1,2]*, Andrew Mugler[1,3]*

[1]Department of Physics and Astronomy, Purdue University, West Lafayette, United States; [2]Department of Physics and Astronomy, University of Pennsylvania, Philadelphia, United States; [3]Department of Physics and Astronomy, University of Pittsburgh, Pittsburgh, United States

**Abstract** Morphogen profiles allow cells to determine their position within a developing organism, but not all morphogen profiles form by the same mechanism. Here, we derive fundamental limits to the precision of morphogen concentration sensing for two canonical mechanisms: the diffusion of morphogen through extracellular space and the direct transport of morphogen from source cell to target cell, for example, via cytonemes. We find that direct transport establishes a morphogen profile without adding noise in the process. Despite this advantage, we find that for sufficiently large values of profile length, the diffusion mechanism is many times more precise due to a higher refresh rate of morphogen molecules. We predict a profile lengthscale below which direct transport is more precise, and above which diffusion is more precise. This prediction is supported by data from a wide variety of morphogens in developing *Drosophila* and zebrafish.

## Introduction

Within developing organisms, morphogen profiles provide cells with information about their position relative to other cells. Cells use this information to determine their position with extremely high precision (*Dubuis et al., 2013*; *Erdmann et al., 2009*; *Gregor et al., 2007a*; *Houchmandzadeh et al., 2002*; *de Lachapelle and Bergmann, 2010*). However, not all morphogen profiles are formed via the same mechanism and for some profiles the mechanism is still not well understood. One well-known mechanism is the synthesis-diffusion-clearance (SDC) model in which morphogen molecules are produced by localized source cells and diffuse through extracellular space before degrading or being internalized by target cells (*Akiyama and Gibson, 2015*; *Gierer and Meinhardt, 1972*; *Lander et al., 2002*; *Müller et al., 2013*; *Rogers and Schier, 2011*; *Wilcockson et al., 2017*). Alternatively, a direct transport (DT) model has been proposed where morphogen molecules travel through protrusions called cytonemes directly from the source cells to the target cells (*Akiyama and Gibson, 2015*; *Bressloff and Kim, 2018*; *Kornberg and Roy, 2014*; *Müller et al., 2013*; *Wilcockson et al., 2017*). The presence of these two alternative theories raises the question of whether there exists a difference in the performance capabilities between cells utilizing one or the other.

Experiments have shown that morphogen profiles display many characteristics consistent with the SDC model. The concentration of morphogen as a function of distance from the source cells has been observed to follow an exponential distribution for a variety of different morphogens (*Driever and Nüsslein-Volhard, 1988*; *Houchmandzadeh et al., 2002*). The accumulation times for several morphogens in *Drosophila* have been measured and found to match the predictions made by the SDC model (*Berezhkovskii et al., 2011*). In zebrafish, the molecular dynamics of the morphogen Fgf8 have been measured and found to be consistent with Brownian diffusion through extracellular space (*Yu et al., 2009*). Despite these consistencies, recent experiments have lent support to

*For correspondence:
sfancher@sas.upenn.edu (SF);
andrew.mugler@pitt.edu (AM)

**Competing interests:** The authors declare that no competing interests exist.

the theory that morphogen molecules are transported through cytonemes rather than extracellular space. The establishment of the Hedgehog morphogen gradient in *Drosophila* is highly correlated in both space and time with the formation of cytonemes (*Bischoff et al., 2013*), while Wnt morphogens have been found to be highly localized around cell protrusions such as cytonemes (*Huang and Kornberg, 2015*; *Stanganello and Scholpp, 2016*). Theoretical studies of both the SDC and DT models have examined these measurable effects (*Berezhkovskii et al., 2011*; *Bressloff and Kim, 2018*; *Shvartsman and Baker, 2012*; *Teimouri and Kolomeisky, 2016a*), but direct comparisons between the two models have thus far been poorly explored. In particular, it remains unknown whether one model allows for a cell to sense its local morphogen concentration more precisely than the other given biological parameters such as the number of cells or the characteristic lengthscale of the profile.

Here, we derive fundamental limits to the precision of morphogen concentration sensing for both the SDC and DT models. We investigate the hypothesis that sensory precision plays a major role in the selection of a gradient formation mechanism during evolution, and we test this hypothesis by quantitatively comparing our theory to morphogen data. Intuitively one might expect the DT model to have less noise due to the fact that molecules are directly deposited at their target. Indeed, we find below that the noise arises only from molecular production and degradation, with no additional noise from molecular transport. However, we also find below that for sufficiently large morphogen profile lengthscales, the SDC model produces less noise than the DT model due to it being able achieve a higher effective unique molecule count. By elucidating the competing effects of profile amplitude, steepness, and noise, we ultimately conclude that there should exist a profile lengthscale below which the DT model is more precise and above which the SDC mechanism is more precise. We find that this prediction is quantitatively supported by data from a wide variety of morphogens, suggesting that readout precision plays an important role in determining the mechanisms of morphogen profile establishment.

## Results

Several past studies have focused on the formation dynamics of morphogen profiles (*Berezhkovskii et al., 2011*; *Bressloff and Kim, 2018*; *Shvartsman and Baker, 2012*; *Teimouri and Kolomeisky, 2016a*). Here, we model profiles in the steady state regime, as most of the experimental measurements to which we will later compare our results were taken during stages when the steady state approximation is valid (*Grimm et al., 2010*; *Gregor et al., 2007b*; *Kicheva et al., 2007*; *Yu et al., 2009*; *Kanodia et al., 2009*). Precision depends not only on stochastic fluctuations in the morphogen concentration, but also on the shape of the mean morphogen profile, as the shape determines concentration differences between adjacent cells that may adopt different fates. Therefore, as in past studies (*Gregor et al., 2007a*; *Tostevin et al., 2007*), we define the precision as $P = \Delta m_j / \sigma_j$, where $\sigma_j$ is the standard deviation of the number of morphogen molecules arriving at cell $j$, and $\Delta m_j = m_j - m_{j+1}$ is the difference between the molecule number in that cell and the adjacent cell. As is typical in studies of both the DT (*Teimouri and Kolomeisky, 2016a*; *Bressloff and Kim, 2018*) and SDC (*Berezhkovskii et al., 2011*; *Shvartsman and Baker, 2012*; *Teimouri and Kolomeisky, 2016b*) mechanisms, we focus on a one-dimensional line of target cells. However, we derive analogous results for 2D and 3D systems, and we generally find that the dimensionality does not qualitatively change our results, as we discuss later. In 1D, cells extend in both directions from the source cell, with $N$ cells on each side (*Figure 1*). We note that in the case of the Bicoid morphogen in the *Drosophila* embryo, target cells extend only on one side of the source. This will introduce a factor of 2 in the means of both the DT and SDC models and a factor slightly greater than 2 in the variance of the SDC model. This will not affect the agreement of the Bicoid data with our theory in Figure 4B.

A common method for determining the statistical properties of stochastic systems is to express the dynamics of their probability distributions in the form of a master equation. The first moment of the master equation then dictates the dynamics of the mean and becomes the rate equation. Higher order moments can be similarly used to calculate other statistical properties such as the variance. As we will show, in the case of the DT mechanism only the rate equation will be needed as the relevant statistical properties are identical to that of a simple birth-death system which is fully characterized by its mean. Conversely, the SDC mechanism has more complicated noise properties, which we will

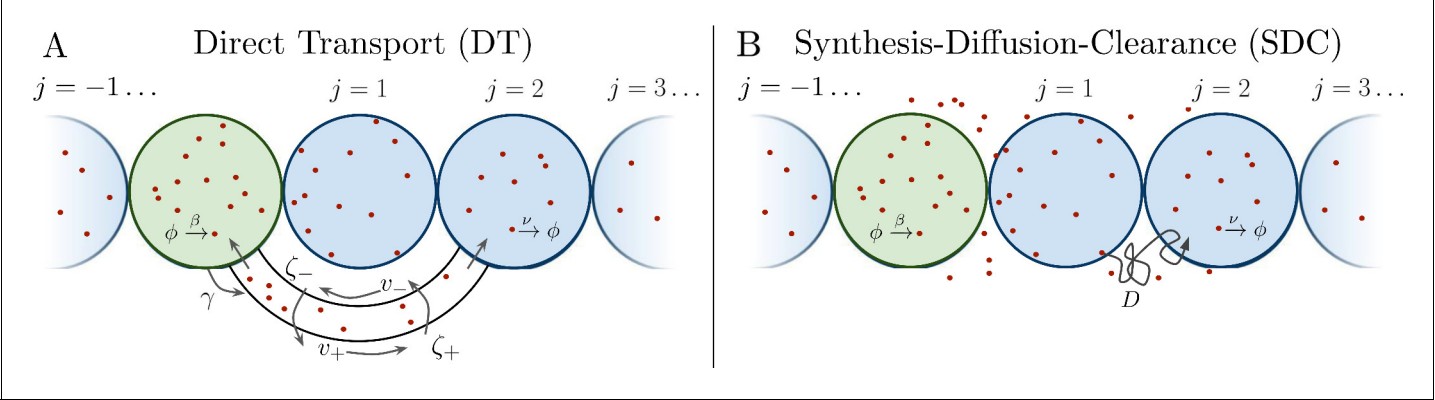

**Figure 1.** Source cell (green) produces morphogen which is delivered to N target cells (blue) on either side via (**A**) direct transport (DT) or (**B**) synthesis-diffusion-clearance (SDC).

calculate via the first and second moments. Specifically, we will use a Langevin description which is expressed as a rate equation with noise terms *Gardiner, 2004*.

## Direct transport model

We first consider the DT case, where morphogen molecules are transported via cytonemes that connect a single source cell to multiple target cells (*Figure 1A*). Cytonemes are tubular protrusions that are hundreds of nanometers thick and between several and hundreds of microns long (*Kornberg and Roy, 2014*; *Kornberg, 2014*). They are supported by actin filaments, and it is thought that morphogen molecules are actively transported along the filaments via molecular motors (*Kornberg and Roy, 2014*; *Kornberg, 2014*; *Sanders et al., 2013*; *Huang and Kornberg, 2015*). It was recently shown that a DT model that includes forward and backward transport of molecules within cytonemes reproduces experimentally measured accumulation times (*Teimouri and Kolomeisky, 2016a*; *Bressloff and Kim, 2018*), although the noise properties of this model were not considered. Here, we review the steady state properties of this model and derive its noise properties.

Consider a single source cell that produces morphogen at rate $\beta$. Morphogen molecules enter each cytoneme at rate $\gamma$. The cytoneme that leads to the $j$th target cell has length $2ja$, where $a$ is the cell radius. Once inside a cytoneme, morphogen molecules move forward toward the target cell with velocity $v^+$ or backwards toward the source cell with velocity $v^-$, and can switch between these states with rates $\zeta^+$ (forward-to-backward) or $\zeta^-$ (backward-to-forward). Once a molecule reaches the forward (backward) end of the cytoneme, it is immediately absorbed into the target (source) cell. Molecules within a target cell spontaneously degrade with rate $\nu$. An alternative model could involve neglecting this degradation step by counting the arrival of morphogen molecules rather than the concentration. This method can at best reduce the variance by a factor of 2 as the noise from degradation is eliminated but the noise from arrival remains. This would cause negligible change to our results presented in Figure 3 and Figure 4 given the order of magnitude difference in precision seen between our two models.

The dynamics of the mean number of morphogen molecules in the source cell $m_0(t)$ and $j$th target cell $m_j(t)$, and the mean density of forward-moving molecules $u_j^+(x,t)$ and backward-moving molecules $u_j^-(x,t)$ in the $j$th cytoneme are (*Bressloff and Kim, 2018*)

$$
\begin{aligned}
\frac{\partial m_0}{\partial t} &= \beta - \sum_j \left[ \gamma m_0 - v^- u_j^-(0,t) \right], \\
\frac{\partial u_j^+}{\partial t} &= -v^+ \frac{\partial u_j^+}{\partial x} + \zeta^- u_j^- - \zeta^+ u_j^+, \\
\frac{\partial u_j^-}{\partial t} &= v^- \frac{\partial u_j^-}{\partial x} - \zeta^- u_j^- + \zeta^+ u_j^+, \\
\frac{\partial m_j}{\partial t} &= v^+ u_j^+(L_j,t) - \nu m_j,
\end{aligned}
\tag{1}
$$

where the summation in the first line runs from $j=-N$ to $j=N$ excluding 0 (the source cell).

Additionally, the boundary conditions $v^+ u_j^+(0,t) = \gamma m_0(t)$ and $v^- u_j^-(L_j,t) = 0$ are imposed to reflect the rate at which morphogen molecules enter the cytoneme from either end. This creates a steady-state solution of the form

$$m_j^{\mathrm{DT}} = \frac{\beta \Gamma_j}{2\nu \sum_{k=1}^N \Gamma_k}, \text{ where } \Gamma_j = \frac{e^{-2|j|\kappa a}(1 - e^{-\phi})}{1 - e^{-\phi - 2|j|\kappa a}}. \tag{2}$$

This solution is identical to that found in *Bressloff and Kim, 2018* with the exception of the factor of $1/2$ that accounts for target cells existing on both sides of the source cell in our model. Here, $\gamma \Gamma_j$ is the effective transport rate of morphogen molecules to the $j$th target cell, and $\phi = \log(d^-/d^+)$ and $\kappa = (d^+)^{-1} - (d^-)^{-1}$ are defined in terms of the average distance a molecule would move forward $d^+ = v^+/\zeta^+$ or backward $d^- = v^-/\zeta^-$ within a cytoneme before switching direction. The parameter $\phi$ sets the shape of $\Gamma_j$, and thus of $m_j$: when $\phi \ll -1$ the profile is constant, $\Gamma_j = 1$; when $\phi \gg 1$ it is exponential, $\Gamma_j = e^{-2|j|\kappa a}$; and when $|\phi| \ll 1$ it is a power law for large $j$, $\Gamma_j = (1 + 2|j|a/d^+)^{-1}$. The parameter $\kappa$ sets the lengthscale of the profile, defined as

$$\lambda_{\mathrm{DT}} = \sum_{j=1}^N \frac{\Gamma_j - \Gamma_N}{\Gamma_1 - \Gamma_N} \approx \frac{1}{|\kappa|}\left(e^{|\phi|} - 1\right)\left(|\phi| - \log\left(e^{|\phi|} - 1\right)\right), \tag{3}$$

where we approximate the sum as an integral for $N \gg 1$. We use this expression to eliminate $\kappa$, writing $\Gamma_j$ in *Equation 2* entirely in terms of $\phi$ and $\hat{\lambda} \equiv \lambda/a$.

Despite the complexity of the transport process in *Equation 1*, we find that it adds no noise to $m_j$. In fact, here we prove that any system in which molecules can only degrade in the target cells and cannot leave the target cells has the steady-state statistical properties of a simple birth-death process. First, consider the special case of only one target cell. Because each morphogen molecule produced in the source cell acts independently of every other morphogen molecule, we define $p(\tau)$ as the probability density that any given molecule will enter the target cell a time $\tau$ after it is created in the source cell. Next, we define $Q(\delta t)$ as the probability that a morphogen molecule will enter the target cell between $t$ and $t + \delta t$. This event requires the molecule to have been produced between $t - \tau$ and $t - (\tau + d\tau)$, which occurs with probability $\beta d\tau$; to arrive at the target cell a time $\tau$ later and to enter the target cell within the window $\delta t$, which occurs with probability $p(\tau)\delta t$; and we must integrate over all possible times $\tau$. Therefore,

$$Q(\delta t) = \int_0^\infty [\beta d\tau][p(\tau)\delta t] = \beta \delta t \int_0^\infty d\tau \, p(\tau) = \beta \delta t, \tag{4}$$

where the last step follows from normalization.

This generalized version of the DT model is visualized in *Figure 2*. We see that regardless of the form of $p(\tau)$, the probability of a morphogen molecule entering the target cell in any given small time window $\delta t$ is simply $\beta \delta t$. Since the mechanism by which morphogen molecules go from the source cell to the target cell can only affect $p(\tau)$, this result holds regardless of the specifics of the mechanism so long as the condition that the morphogen cannot leave the target cell other than by degradation is maintained. This result also holds when the system is expanded to have multiple target cells, as then $p(\tau)$ is replaced with $p_j(\tau)$, the probability density that the molecule enters the $j$th target cell a time $\tau$ after being produced. In this case, $\int_0^\infty d\tau \, p_j(\tau)$ evaluates to $\pi_j$, the total probability the morphogen molecule is ultimately transported to the $j$th target cell, and $\beta \delta t$ is simply replaced with $\beta \pi_j \delta t$. Combined with the constant degradation rate $\nu$ of morphogen molecules within the target cell, this is precisely a birth-death process with birth rate $\beta \pi_j$ and death rate $\nu$. For our system $\pi_j = \Gamma_j/2\sum_{k=1}^N \Gamma_k$ in *Equation 2*. The conditions that morphogen molecules act independently and can only degrade within the target cell are critical as the former ensures the noise sources are linear and the later will be violated by the SDC mechanism.

We now assume that each cell integrates its morphogen molecule count over a time $T$ (*Berg and Purcell, 1977*; *Gregor et al., 2007a*). The variance in the time average $T^{-1}\int_0^T dt > m_j(t)$ is simply that of a birth-death process, given by $\sigma_j^2 = 2m_j/(T/\tau)$ (*Fancher and Mugler, 2017*), so long as $T \gg \tau$, where $\tau = \nu^{-1}$ is the correlation time. We see that, as expected for a time-averaged Poisson

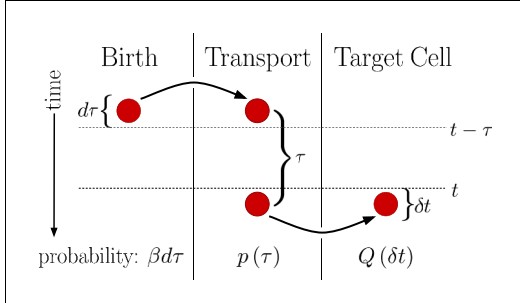

**Figure 2.** Diagram outlining a generalized version of the DT model. Between times $t - (\tau + d\tau)$ and $t - \tau$ a molecule is produced with probability $\beta d\tau$. This molecule then undergoes a transport process over a time $\tau$ with probability density $p(\tau)$. Finally, the molecule is deposited into the target cell at the end of the transport process. Integrating over all possible values of $\tau$ then yields $Q(\delta t)$, the probability of the molecule entering the target cell between times $t$ and $t + \delta t$.

process, the variance increases with the mean $m_j$ and decreases with the number $T/\tau$ of independent measurements made in the time $T$. The precision is therefore

$$P_{\text{DT}}^2 = \frac{m_j^{\text{DT}} T}{2\tau_{\text{DT}}} \left(\frac{\Delta m_j}{m_j^{\text{DT}}}\right)^2, \text{ with } \tau_{\text{DT}} = \frac{1}{\nu}. \quad (5)$$

We see that the precision increases with the profile amplitude $m_j$, the number of independent measurements $T/\tau$, and the profile steepness $\Delta m_j/m_j$. The transport process influences the precision only via $m_j$, not $\tau$. For a given $N$, $j$, and $\hat{\lambda}$, we find that the precision is maximized at a particular $\phi^* > 0$ (**Figure 3A**). The reason is that an exponential profile ($\phi \gg 1$) has constant steepness but small amplitude, whereas a power-law profile ($\phi \ll 1$) has low steepness but large amplitude due to its long tail; the optimum is in between.

## SDC model

We next consider the SDC case (**Figure 1B**). Again a single source cell at the origin $x = 0$ produces morphogen at rate $\beta$. However, now morphogen molecules diffuse freely along $x$ with coefficient $D$ and degrade spontaneously at any point in space with rate $\nu$. The dynamics of the morphogen concentration $c(x, t)$ are

$$\frac{\partial c}{\partial t} = D\nabla^2 c + \eta_D - \nu c - \eta_\nu + \left(\beta + \eta_\beta\right)\delta(x), \quad (6)$$

where the noise terms associated with diffusion, degradation, and production obey

$$\begin{aligned}
\langle \eta_D(x', t')\eta_D(x, t)\rangle &= 2D\delta(t - t')\vec{\nabla}_x \cdot \vec{\nabla}_{x'} c(x)\delta(x - x') \\
\langle \eta_\nu(x', t')\eta_\nu(x, t)\rangle &= \nu c(x)\delta(t - t')\delta(x - x'), \\
\langle \eta_\beta(t')\eta_\beta(t)\rangle &= \beta\delta(t - t'),
\end{aligned} \quad (7)$$

respectively (**Gardiner, 2004**; **Gillespie, 2000**; **Fancher and Mugler, 2017**; **Varennes et al., 2017**). Here, the time independent $c(x) = \beta e^{-|x|/\lambda}/(2\nu\lambda)$ is the steady state mean concentration, with characteristic lengthscale $\lambda_{\text{SDC}} = \sqrt{D/\nu}$. We imagine a target cell located at $x$ that is permeable to the morphogen and counts the number $m(x, t) = \int_V dy\, c(x + y, t)$ of morphogen molecules within its volume $V$. We use this simpler prescription over explicitly accounting for more realistic mechanisms such as surface receptor binding because it has been shown that the two approaches ultimately yield similar concentration sensing results up to a factor of order unity (**Berg and Purcell, 1977**). For a cell in steady state at position $x = 2ja$, the integral evaluates to

$$m_j^{\text{SDC}} = (\beta/\nu)\sinh(1/\hat{\lambda})e^{-2|j|/\hat{\lambda}}. \quad (8)$$

Importantly, in the model presented here the morphogen molecules can diffuse both into and out of the target cells, thus violating the condition that they can only degrade once in a target cell and disallowing our previous argument depicted in **Figure 2**. However, because **Equation 6** is linear with Gaussian white noise, calculating the time-averaged variance $\sigma_j^2$ is straightforward: we Fourier transform **Equation 6** in space and time, calculate the power spectrum of $m(x, t)$, and take its low-frequency limit (Appendix 1). So long as $T \gg \nu^{-1}$, we obtain the same functional form as **Equation 5**,

$$P_{\text{SDC}}^2 = \frac{m_j^{\text{SDC}} T}{2\tau_{\text{SDC}}} \left(\frac{\Delta m_j^{\text{SDC}}}{m_j^{\text{SDC}}}\right)^2, \quad (9)$$

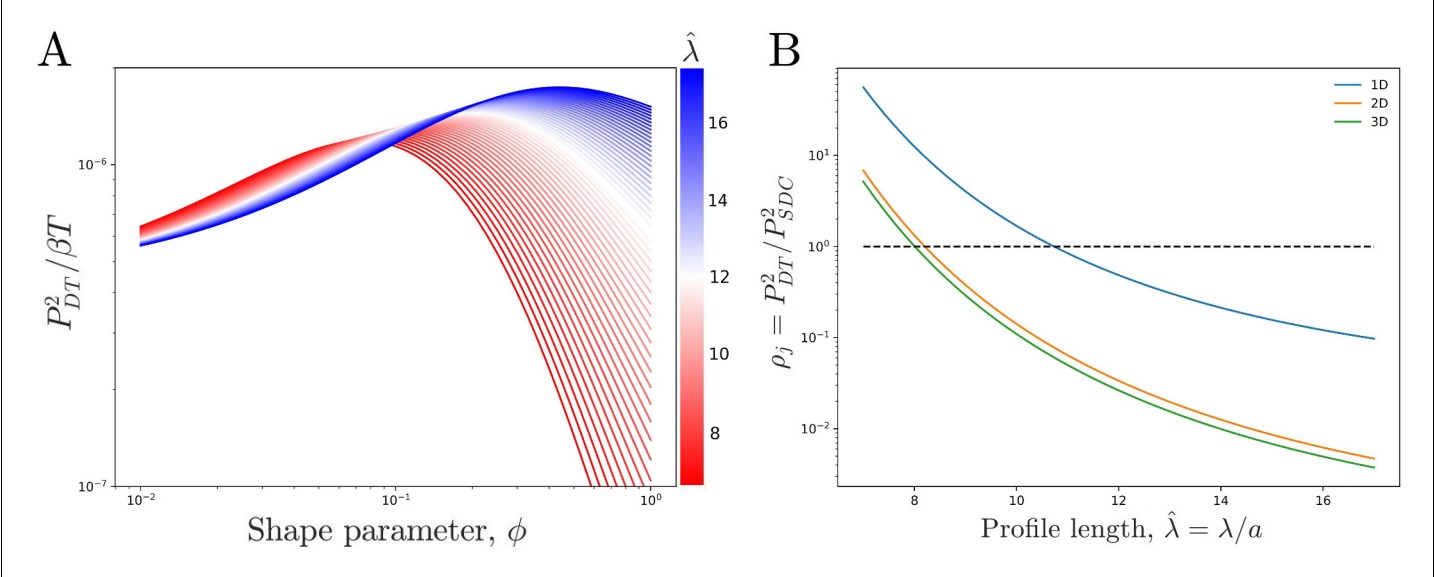

**Figure 3.** Comparing theoretical DT precision to SDC precision for a single cell. (A) DT precision shows a maximum as a function of shape parameter $\phi$ for any value of the profile lengthscale. (B) Ratio $\rho_j$ of DT to SDC precision shows a crossover ($\rho_j = 1$) as a function of profile lengthscale $\lambda/a$ for 1D, 2D, and 3D geometries. Here $j = 50$ is the central cell of $N = 100$ target cells. For each value of $\hat{\lambda}$ the value of $\phi$ which maximizes precision in the DT model ($\phi^*$) as seen in A is used.

because diffusion is a Poisson process. This result is distinguished from that of the DT system by the correlation time taking the form

$$\tau_{\text{SDC}} = \frac{1}{\nu}\left[1 - \frac{(2/\hat{\lambda}) + \sinh(2/\hat{\lambda})}{4\sinh(1/\hat{\lambda})e^{1/\hat{\lambda}}}\right]. \tag{10}$$

The factor in brackets is always less than one and decreases with $\hat{\lambda}$. It reflects the fact that, unlike in the DT model, molecules can leave a target cell not only by degradation, but also by diffusion. Therefore, the rate $\tau^{-1}$ at which molecules are refreshed is larger than that from degradation alone. This effect increases the precision because more independent measurements ($T/\tau$) can be made.

To understand this effect more intuitively, consider a simplified SDC model in which diffusion is modeled as discrete hopping between adjacent target cells at rate $h$. The autocorrelation function is $C_j(t) = m_j I_0(2ht)e^{-(2h+\nu)t}$ (Appendix 2), where $I_0$ is the zeroth modified Bessel function of the first kind. The correlation time is $\tau = \int_0^\infty dt\, C_j(t)/C_j(0) = [\nu(4h + \nu)]^{-1/2}$, and we see explicitly that it decreases with both degradation ($\nu$) and diffusion ($h$). In fact, in the limit of fast diffusion ($h \gg \nu$), the expression becomes $\tau = (4\nu h)^{-1/2}$. Correspondingly, in the fast-diffusion limit of *Equation 9* ($\hat{\lambda} \gg 1$), the term in brackets reduces to $\hat{\lambda}^{-1}$, and it becomes $\tau = (\nu\hat{\lambda})^{-1/2} = [4\nu D/(2a)^2]^{-1/2}$. These expressions are identical, with $D/(2a)^2$ playing the role of the hopping rate $h$, as expected.

## Comparing the models

We now ask which model has the higher precision. We first note that while the precision in both models has the same form when expressed in terms of the means and correlation times (*Equations 5 and 8*), substituting in the corresponding expressions reveals how the precision depends on the various parameters of each model. Specifically, the precision in the DT model is seen to depend on $N$, $j$, the product $\beta T$, $\hat{\lambda}$, and $\phi$. Importantly, it is independent of $\nu$ so long as the condition $T \gg \tau_{\text{DT}} = \nu^{-1}$ is met. This is due to the mean molecule count scaling as $\nu^{-1}$ (*Equation 2*) and the number of independent measurements ($T/\tau_{\text{DT}} = \nu T$) scaling as $\nu$, thus causing their product and in turn the precision to be independent of $\nu$. In the SDC model, the precision depends on $j$, the product $\beta T$, and $\hat{\lambda}$. For similar reasons as in the DT model, $\nu$ does not explicitly appear once *Equations 7 and 9* are

inserted into *Equation 8*, but it is present implicitly in the definition of $\hat{\lambda} = \sqrt{D/\nu}/a$.

To properly compare the two models, we equate several variables. Specifically, we wish to compare the precision of each model within a specific system, which requires $N$ and $j$ to be the same in both models. Additionally, $\beta$ and $\nu$ are restricted by the energy and material costs of producing and degrading morphogen molecules while $T$ is restricted by the need of the system to properly develop in a finite amount of time. These restrictions are assumed to be independent of the method of morphogen profile establishment, which implies that both the DT and SDC systems will adopt the same maximal values of $\beta$ and $T$. While the value of $\nu$ is restricted in a similar manner, explicitly equating it between the two models does not reduce the number of free parameters, as the precision is independent of $\nu$ in the DT model, and $\nu$ is insufficient to fully define the value of $\hat{\lambda}$ in the SDC model. Therefore, we equate the characteristic profile length scale $\hat{\lambda}$, as this parameter is consistently defined across both models and is also measured in experiments, allowing us to compare our theory with data as discussed below. Finally, we optimize over $\phi$ as it is unique to the DT model.

We now observe how the precision of the DT and SDC models compare in a representative system. *Figure 3B* shows $\rho_j = P_{\mathrm{DT}}^2/P_{\mathrm{SDC}}^2$ as a function of profile length $\hat{\lambda}$ for a cell in the center ($j = N/2$) of a line of $N = 100$ target cells, where for each $\hat{\lambda}$ we use the $\phi^*$ that maximizes $P_{\mathrm{DT}}^2$ as seen in *Figure 3A*. Since $\beta$ and $T$ have been equated between the two models, $\rho_j$ is independent of both. We see that for short profiles the DT model is more precise ($\rho_j>1$) whereas for long profiles the SDC model is more precise ($\rho_j<1$). This effect holds for a single source cell providing morphogen for a 1D line of target cells as well as for a 1D line of source cells with a 2D sheet of target cells and a 2D sheet of source cells with a 3D volume of target cells. For the DT model, the 2D and 3D cases are identical to the 1D case as we assume that cytonemes extend perpendicular to the source cells; for the SDC model see Appendix 1.

The reason that the SDC model is more precise for long profiles is that long profiles correspond to fast diffusion, which increases the refresh rate $\tau_{\mathrm{SDC}}^{-1}$ as discussed above. Conversely, the reason that the DT model is more precise for short profiles is that it has a larger amplitude. It also has a smaller steepness, but the larger amplitude wins out. Specifically, whereas the SDC amplitude falls off exponentially, $m_j \sim e^{-2|j|/\hat{\lambda}}$, for sufficiently small $\phi^*$ the DT amplitude falls off as a power law, $m_j \sim 1/|j|$. The steepness $\Delta m_j/m_j$ of the SDC profile is constant, while the steepness of the DT profile also scales like $1/j$. Thus, the product of the ratio of amplitudes and the square of the ratio of steepnesses, on which $\rho_j$ depends, scales like $e^{2|j|/\hat{\lambda}}/|j|^3$. For small $\hat{\lambda}$, the exponential dominates over the cubic for the majority of $j$ values. Consequently, the DT model has the higher precision.

## Comparison to data

We now test our predictions against data for various morphogens. In *Drosophila*, the morphogen Wingless (Wg) is localized near cell protrusions such as cytonemes (*Huang and Kornberg, 2015*; *Stanganello and Scholpp, 2016*), and the Hedgehog (Hh) gradient correlates highly in both space and time with the formation of cytonemes (*Bischoff et al., 2013*), suggesting that these two morphogen profiles are formed via a DT mechanism. Conversely, Bicoid has been understood as a model example of SDC for decades (*Driever and Nüsslein-Volhard, 1988*; *Gregor et al., 2007a*; *Houchmandzadeh et al., 2002*). Similarly, Dorsal is spread by diffusion; however, its absorption is localized to a specific region of target cells via a nonuniform degradation mechanism, making it more complex than the simple SDC model (*Carrell et al., 2017*). Finally, for Dpp there is evidence for a variety of different gradient formation mechanisms (*Akiyama and Gibson, 2015*; *Müller et al., 2013*; *Wilcockson et al., 2017*).

In zebrafish, the morphogen Fgf8 has been studied at the single molecule level and found to have molecular dynamics closely matching the Brownian movement expected in an SDC mechanism (*Yu et al., 2009*). Similarly, Cyclops, Squint, Lefty1, and Lefty2, all of which are involved in the Nodal/Lefty system, have been shown to spread diffusively and affect cells distant from their source (*Müller et al., 2013*; *Rogers and Müller, 2019*). This would support the SDC mechanism, although Cyclops and Squint have been argued to be tightly regulated via a Gierer-Meinhardt type system, thus diminishing their gradient sizes to values much lower than what they would be without this regulation (*Gierer and Meinhardt, 1972*; *Rogers and Müller, 2019*).

For all these morphogens, we estimate the profile lengthscales $\lambda$ from the experimental data (*Kicheva et al., 2007*; *Wartlick et al., 2011*; *Gregor et al., 2007b*; *Liberman et al., 2009*; *Yu et al., 2009*; *Müller et al., 2012*) (Appendix 3). *Figure 4A* shows these $\lambda$ values and indicates for each morphogen whether the evidence described above suggests a DT mechanism (red), an SDC mechanism (blue), or multiple mechanisms including DT and SDC (white). We see that in general, the three cases correspond to short, long, and intermediate profile lengths, respectively, which is qualitatively consistent with our predictions.

To make the comparison quantitative, we estimate the values of cell radius $a$ and cell number $N$ from the experimental data (*Kicheva et al., 2007*; *Gregor et al., 2007a*; *Liberman et al., 2009*; *Yu et al., 2009*; *Kimmel et al., 1995*) (Appendix 3) in order to calculate $\rho_j$ from our theory in each case. The background color in Fig. *Figure 4B* shows the percentage of cells within each system for which we predict that the SDC model is more precise as a function of $\hat{\lambda}$. The data points in Fig. *Figure 4B* show the values of $\hat{\lambda}$ from the experiments, also normalized by $\hat{\lambda}_{50}$, the value of $\hat{\lambda}$ at which 50% of cells are more precise in SDC, from the theory. For each morphogen species, we assume a 1D system for simplicity as we have checked that considering higher dimensions yields negligible differences to the results presented in *Figure 4B*. We see that our theory predicts the correct threshold: the morphogens for which the evidence suggests either a DT or an SDC mechanism (red or blue) fall into the regime in which we predict that mechanism to be more precise for most of the cells, and the morphogens with multiple mechanisms (white) fall in between. This result provides quantitative support for the idea that morphogen profiles form according to the mechanism that maximizes the sensory precision of the target cells.

## Discussion

We have shown that in the steady-state regime, the DT and SDC models of morphogen profile formation yield different scalings of readout precision with the length of the profile and population size. As a result, there exist regimes in this parameter space in which either mechanism is more precise. While the DT model benefits from larger molecule numbers and no added noise from the transport process, the ability of molecules to diffuse into and away from a target cell in the SDC model allows the cell to measure a greater number of effectively unique molecules in the same time frame. By examining how these phenomena affect the cells' sensory precision, we predicted that morphogen profiles with shorter lengths should utilize cytonemes or some other form of direct transport mechanism, whereas morphogens with longer profiles should rely on extracellular diffusion, a

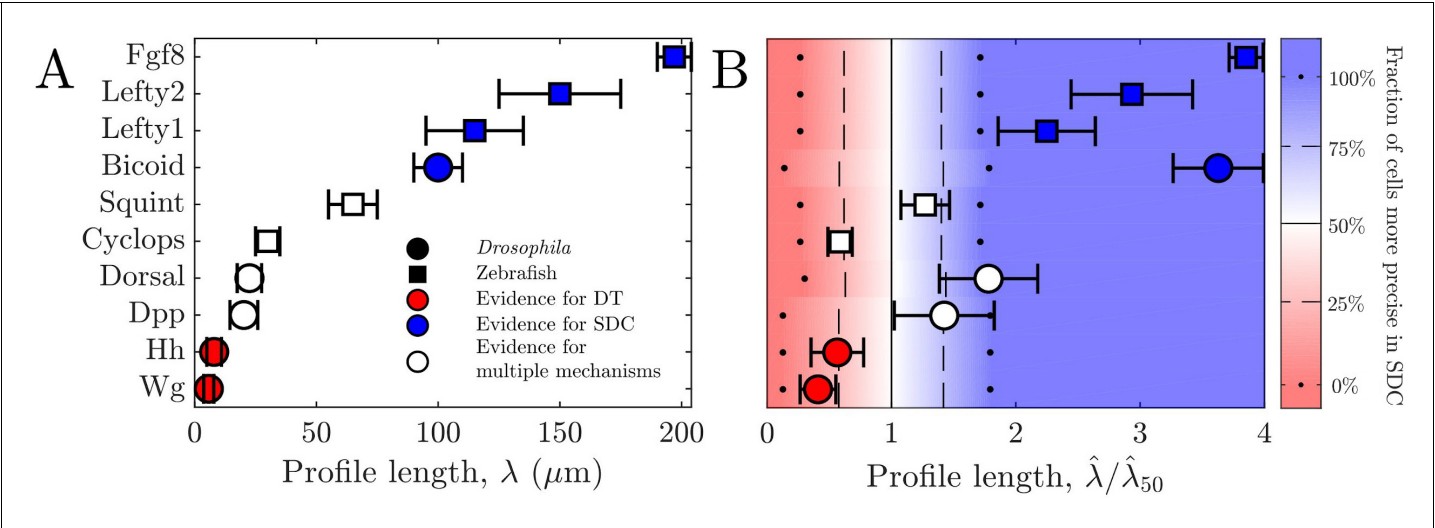

**Figure 4.** Comparing theory and experiment. (**A**) $\lambda$ values for morphogens estimated from experiments, colored by whether experiments support a DT (red), SDC (blue), or multiple mechanisms (white). (**B**) Data from A overlaid with color from theory using values of $a$ and $N$ estimated from experiments. Color indicates percentage of cells for which SDC is predicted to be more precise with dots signifying 0% and 100%, dashed lines signifying 25% and 75%, and solid lines signifying 50%. The $\hat{\lambda}$ axis is normalized by $\hat{\lambda}_{50}$, the value of $\hat{\lambda}$ at which 50% of cells are more precise in SDC.

prediction that is in quantitative agreement with measurements on known morphogens. It will be interesting to observe whether this trend is further strengthened as more experimental evidence is obtained for different morphogens, as well as to expand the theory of multicellular concentration sensing to further biological contexts.

Despite the quantitative agreement between our theory and experiments, it is clear that the models presented here are minimal and thus cannot be directly applied to all systems. This is exemplified by morphogen such as Dorsal, which due to aforementioned diffusive spreading and nonuniform degradation mechanism clearly does not strictly follow either model. Additionally, the SDC model can be violated if the diffusion of morphogen through a biological environment is hindered by the typically crowded nature of such environments, leading to possibly subdiffusive behavior (*Ellery et al., 2014*; *Fanelli and McKane, 2010*). For the DT model, we explicitly ignored the dynamics of the cytonemes themselves due to the growth rate of the cytonemes being sufficiently fast so as to traverse the entire system size in significantly less time than is required for the cells to integrate their morphogen counts over (*Bischoff et al., 2013*; *Chen et al., 2017*). This assumption is problematic if cytonemes continue to behave dynamically after reaching the source cell. In particular, the process of cytonemes switching between phases of growing and retracting can introduce super-Poissonian noise sources to the morphogen count within the target cells. Super-Poissonian noise can also be introduced by relaxing the assumption that the morphogen molecules behave independently of each other as this was a critical component of our proof that the molecule count in the target cells is statistically identical to a birth-death process. Finally, it is conceivable that a hybrid system which utilizes a combination of diffusion and directed transport could be developed. However, we are unaware of any experimental study into such a possibility and as such have not considered it in this particular work. It will be interesting to explore the implications of each of these complications in future works.

## Materials and methods

Methods are described in Appendices 1, 2, and 3 along with more detailed evaluations of 2D and 3D SDC systems and further discussion of data used in *Figure 4*. Additionally, code used to generate the plots seen in *Figure 3* and *Figure 4* can be found at *Fancher, 2020*.

## Acknowledgements

This work was supported by Simons Foundation Grants No. 376198 and 568888 and National Science Foundation Grant No. PHY-1945018. We thank Chris Bairnsfather for useful discussions.

## Additional information

### Funding

| Funder | Grant reference number | Author |
| --- | --- | --- |
| Simons Foundation | 376198 | Sean Fancher Andrew Mugler |
| Simons Foundation | 568888 | Sean Fancher |
| National Science Foundation | PHY-1945018 | Andrew Mugler |

The funders had no role in study design, data collection and interpretation, or the decision to submit the work for publication.

### Author contributions

Sean Fancher, Conceptualization, Data curation, Software, Formal analysis, Investigation, Visualization, Methodology, Writing - original draft, Writing - review and editing; Andrew Mugler, Conceptualization, Supervision, Funding acquisition, Visualization, Methodology, Writing - original draft, Project administration, Writing - review and editing

## Author ORCIDs

Sean Fancher https://orcid.org/0000-0002-8701-192X
Andrew Mugler https://orcid.org/0000-0001-9367-7026

## Decision letter and Author response

Decision letter https://doi.org/10.7554/eLife.58981.sa1
Author response https://doi.org/10.7554/eLife.58981.sa2

## Additional files

### Supplementary files

- Transparent reporting form

### Data availability

All data used in this study is simulated via computational methods outlined in the manuscript and appendices.

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

# Appendix 1

## Time-averaged variance in the SDC model

Here we calculate the time-averaged variance of the morphogen molecule number using the low-frequency limit of the power spectrum. We first introduce the power spectrum, and then we calculate the variance for the 1D, 2D, and 3D geometries.

### Power spectrum

We first discuss the correlation function and power spectrum to establish some definitions and notation. Specifically, we show that the variance in the long-time average of a variable is given by the low-frequency limit of its power spectrum. For a one-dimensional function $x(t)$ with mean 0, the correlation function $C(t)$ takes the form

$$C(t - t') = \langle x(t')x(t) \rangle. \tag{11}$$

Since absolute time is irrelevant in the steady state of any physical system with no time dependent forcing, $t'$ can be set to 0 without loss of generality. This leads to a definition for the power spectrum of $x(t)$ as

$$
\begin{aligned}
S(\omega) &= \int \frac{d\omega'}{2\pi} \langle \tilde{x}^*(\omega')\tilde{x}(\omega) \rangle = \frac{1}{2\pi} \int d\omega' dt dt' \langle x(t')x(t) \rangle e^{i\omega t} e^{-i\omega' t'} \\
&= \int dt dt' C(t - t') e^{i\omega t} \delta(t') = \int dt C(t) e^{i\omega t}.
\end{aligned}
\tag{12}
$$

Thus, under this definition the power spectrum is seen to be the Fourier transform of the correlation function. Additionally, when $x(t)$ is averaged over a time $T$, the time averaged correlation function of $x(t)$ takes the form

$$
\begin{aligned}
C_T(t - t') &= \left\langle \left( \frac{1}{T} \int_{t'}^{t'+T} d\tau' x(\tau') \right) \left( \frac{1}{T} \int_{t}^{t+T} d\tau x(\tau) \right) \right\rangle \\
&= \frac{1}{T^2} \int_{t}^{t+T} d\tau \int_{t'}^{t'+T} d\tau' \langle x(\tau')x(\tau) \rangle \\
&= \frac{1}{T^2} \int_{t}^{t+T} d\tau \int_{t'}^{t'+T} d\tau' C(\tau - \tau').
\end{aligned}
\tag{13}
$$

Let $y \equiv (\tau - \tau') - (t - t')$ and $z \equiv (\tau + \tau') - (t + t')$. This transforms *Equation 13* into

$$
\begin{aligned}
C_T(t - t') &= \frac{1}{T^2} \int_{-T}^{T} dy \int_{|y|}^{2T - |y|} dz \frac{1}{2} C(y + t - t') \\
&= \frac{1}{T^2} \int_{-T}^{T} dy (T - |y|) C(y + t - t').
\end{aligned}
\tag{14}
$$

By inverting the relationship found in *Equation 12*, $C(y + t - t')$ can be replaced with an inverse Fourier transform of $S(\omega)$ to produce

$$
\begin{aligned}
C_T(t - t') &= \frac{1}{T^2} \int_{-T}^{T} dy \int \frac{d\omega}{2\pi} (T - |y|) S(\omega) e^{-i\omega(y + t - t')} \\
&= \int \frac{d\omega}{2\pi} \left( \frac{2}{\omega T} \sin\left( \frac{\omega T}{2} \right) \right)^2 S(\omega) e^{-i\omega(t - t')}.
\end{aligned}
\tag{15}
$$

The factor of $(\omega T)^{-2}$ in the integrand of *Equation 15* forces only small values of $\omega$ to contribute when $T$ is large. Thus, the approximation $S(\omega) \approx S(0)$ can be made since only values of $\omega$ near 0 are contributing. This causes $C_T(0)$, which we will denote as $\sigma^2$ through this and the main text, to be exactly calculable to

$$\sigma^2 = C_T(0) \approx S(0) \int \frac{d\omega}{2\pi} \left( \frac{2}{\omega T} \sin\left( \frac{\omega T}{2} \right) \right)^2 = \frac{S(0)}{T}. \tag{16}$$

Of important note is the fact that this approximation only works if $S(\omega)$ varies slowly compared to $(2\sin(\omega T/2)/\omega T)^2$ near $\omega = 0$. Since $C(t)$ must be time symmetric, $S(\omega)$ must also be symmetric and thus an even function of $\omega$. Thus, near $\omega = 0$ the lowest order correction term for each function will be the second order term. Normalizing each term by the 0-frequency value of each function then lets us to impose the condition

$$\left| \frac{1}{S(0)} \frac{\partial^2 S(\omega)}{\partial \omega^2} \right|_{\omega=0} \ll \left| \frac{\partial^2}{\partial \omega^2} \left( \frac{2}{\omega T} \sin\left( \frac{\omega T}{2} \right) \right)^2 \right|_{\omega=0} = \frac{T^2}{6}. \tag{17}$$

So long as this condition is satisfied, the approximation given in *Equation 16* is valid.

We now cast *Equation 16* into a more intuitive form by considering the correlation time $\tau$, which can be defined as

$$\tau = \int_0^\infty dt \frac{C(t)}{C(0)}. \tag{18}$$

Continuing the use the fact that $C(t)$ must be time symmetric and thus an even function of $t$, *Equation 12* can be used to produce the result

$$S(0) = \int dt C(t) = 2 \int_0^\infty dt C(t) = 2\tau C(0). \tag{19}$$

Inserting this result into *Equation 16* produces

$$\sigma^2 \approx \frac{2\tau}{T} C(0), \tag{20}$$

thus relating the long-time averaged variance, $\sigma^2$, to the instantaneous variance, $C(0)$, and the number of correlation times the system averages over, $T/\tau$.

## Variance and precision

We now consider a model for the Synthesis-Diffusion-Clearance system. We still assume there is a single source cell which produces morphogen at rate $\beta$, but now the morphogen is released into the extracellular environment where it freely diffuses at rate $D$. The morphogen can also spontaneously degrade at rate $\nu$. Even though in the main text we focus on a zero-dimensional source in a one-dimensional space, here we will look at diffusion in a multitude of different spaces with different dimensions as well as morphogen sources that span a multitude of different dimensions. In each case, the sources will secrete morphogen molecules into a density field $c$ which must follow

$$\frac{\partial c}{\partial t} = D\nabla^2 c + \eta_D - \nu c - \eta_\nu + \left( \beta + \eta_\beta \right) \delta^{SP-SO}(\vec{x}), \tag{21}$$

where $SP$ is the number of spatial dimensions, $SO$ is the dimensionality of the source, and $\nabla^2$ is taken over all $SP$ dimensions. Each $\eta$ term is a Langevin noise term that represents Gaussian white noise for the diffusion, degradation, and production processes respectively. Of important note is that $\delta^{SP-SO}(\vec{x})$ is a $\delta$ function only in the last $SP - SO$ dimensions of the space. So, for example, if there was a one dimensional source in three dimensional space, then $\delta^{3-1}(\vec{x})$ would be a $\delta$ function in the $\hat{y}$ and $\hat{z}$ directions but not the $\hat{x}$ direction. This means that $\beta$ and $\eta_\beta$ will have units of $T^{-1}L^{-SO}$, where $T$ is time and $L$ is space.

We can now assume $c$ has reached a steady state and separate it into $c = \bar{c} + \delta c$, which in turn allows *Equation 21* to separate into

$$0 = D\nabla^2 \bar{c} - \nu \bar{c} + \beta \delta^{SP-SO}(\vec{x}) \tag{22}$$

$$\frac{\partial \delta c}{\partial t} = D\nabla^2 \delta c + \eta_D - \nu \delta c - \eta_\nu + \eta_\beta \delta^{SP-SO}(\vec{x}). \tag{23}$$

where

Fourier transforming *Equation 22* in space and dividing it by $\nu$ then yields

$$0 = -\lambda^2 \left|\vec{k}\right|^2 \tilde{\bar{c}} - \tilde{\bar{c}} + \frac{\beta}{\nu}(2\pi)^{SO}\delta^{SO}\left(\vec{k}\right) \implies \tilde{\bar{c}} = \frac{\beta}{\nu}\frac{(2\pi)^{SO}\delta^{SO}\left(\vec{k}\right)}{1+\lambda^2\left|\vec{k}\right|^2},$$ (24)

where

$$\lambda = \sqrt{\frac{D}{\nu}}.$$ (25)

Of similarly important note is that $\delta^{SO}\left(\vec{k}\right)$ is a $\delta$ function only in the **first** $SO$ dimensions of $k$-space. So in the one-dimensional source, three-dimensional space example $\delta^{SO}\left(\vec{k}\right)$ would be a $\delta$ function in the $\hat{x}$ direction of $k$-space but not the $\hat{y}$ or $\hat{z}$ directions.

This allows $\bar{c}$ to be written as

$$\begin{aligned}\bar{c}(\vec{x}) &= \int \frac{d^{SP}k}{(2\pi)^{SP}} e^{-i\vec{k}\cdot\vec{x}}\tilde{\bar{c}}\left(\vec{k}\right) = \frac{\beta}{\nu}\int \frac{d^{SP}k}{(2\pi)^{SP}} e^{-i\vec{k}\cdot\vec{x}}\frac{(2\pi)^{SO}\delta^{SO}\left(\vec{k}\right)}{1+\lambda^2\left|\vec{k}\right|^2}\\ &= \frac{\beta}{\nu}\int \frac{d^{SP-SO}k}{(2\pi)^{SP-SO}} e^{-i\vec{k}\cdot\vec{x}}\frac{1}{1+\lambda^2\left|\vec{k}\right|^2} = \frac{\beta\lambda^{-(SP-SO)}}{\nu}P_{SP-SO}\left(\frac{|\vec{x}|}{\lambda}\right),\end{aligned}$$ (26)

where

$$P_N(x) = \int \frac{d^N u}{(2\pi)^N} e^{-i\vec{u}\cdot\vec{x}}\frac{1}{1+|\vec{u}|^2}.$$ (27)

It is important to note that $P_N$ does not integrate over all available dimensions, but only over the last $N$ dimensions of the space. This in turn means that its argument can only depend on the last $N$ dimensions of any input vector. Returning to the one dimensional source, three dimensional space example, $P_{3-1}(|\vec{x}|/\lambda)$ should only take the $y$ and $z$ components of $\vec{x}$ into account. The $x$ component is made irrelevant by the translational symmetry of the system along the $x$-axis.

Moving on to the noise terms, *Equation 23* can be Fourier transformed in space and time to yield

$$-i\omega\tilde{\delta c} = -D\left|\vec{k}\right|^2\tilde{\delta c} + \tilde{\eta}_D - \nu\tilde{\delta c} - \tilde{\eta}_\nu + \tilde{\eta}_\beta \implies \tilde{\delta c} = \frac{\tilde{\eta}_D - \tilde{\eta}_\nu + \tilde{\eta}_\beta}{\nu\left(1+\lambda^2\left|\vec{k}\right|^2 - i\frac{\omega}{\nu}\right)},$$ (28)

where $\eta_\beta\left(\vec{k},\omega\right)$ depends only on the first $SO$ dimensions of $k$-space. Assuming the $\eta$ terms are all independent of each other allows the cross spectrum of $c$ to be

$$\begin{aligned}\left\langle \tilde{\delta c}^*\left(\vec{k}',\omega'\right)\tilde{\delta c}\left(\vec{k},\omega\right)\right\rangle &= \frac{1}{\nu^2\left(1+\lambda^2\left|\vec{k}\right|^2 - i\frac{\omega}{\nu}\right)\left(1+\lambda^2\left|\vec{k}'\right|^2 + i\frac{\omega'}{\nu}\right)}\\ &\cdot\left(\left\langle \tilde{\eta}_D^*\left(\vec{k}',\omega'\right)\tilde{\eta}_D\left(\vec{k},\omega\right)\right\rangle + \left\langle \tilde{\eta}_\nu^*\left(\vec{k}',\omega'\right)\tilde{\eta}_\nu\left(\vec{k},\omega\right)\right\rangle + \left\langle \tilde{\eta}_\beta^*\left(\vec{k}',\omega'\right)\tilde{\eta}_\beta\left(\vec{k},\omega\right)\right\rangle\right).\end{aligned}$$ (29)

The cross spectrum of $\eta_D$ can be obtained from its correlation function. To derive such a correlation function, we first consider a separate Markovian system comprised of a 1-dimensional lattice of discrete compartments that a diffusing species $Y$ can exist in. The dimensionality is chosen purely for simplicity, as the method outlined below can be easily generalized to higher dimensions to produce the same result. Let $y_i(t)$ be the number of $Y$ molecules in the $i$th compartment at time $t$ and $d$ be the rate at which these molecules move to the $i-1$ or $i+1$ compartment. Given a sufficiently small time step $\delta t$, the probability of a molecule moving from the $i$th compartment to the $i\pm 1$ compartment is

$$P(\{y_i(t+\delta t), y_{i1}(t+\delta t)\} = \{y_i(t)-1, y_{i\pm1}(t)+1\}) = y_i(t)d\delta t. \tag{30}$$

Higher order interactions in which multiple molecules are transfered within the time step $\delta t$ will have probabilites of order $(\delta t)^2$ or higher and can thus be ignored. This allows the mean of $\delta y_i(t) = y_i(t+\delta t) - y_i(t)$ to take the form

$$\langle \delta y_i(t) \rangle = (y_{i-1}(t) + y_{i+1}(t) - 2y_i(t))d\delta t, \tag{31}$$

where the first two terms come from molecules moving into the $i$th compartment from the $i-1$ and $i+1$ compartments respectively and the third term comes from the two different ways molecules can leave the $i$th compartment. As $\delta t$ is small, each of these transfer processes can be treated as being Poissonianly distributed. This allows the variance of $\delta y_i(t)$ to simply be the right-hand side of *Equation 24* but with each term taken to be its absolute value so there are no subtractions. Additionally, this approximation allows the covariance between $\delta y_i$ and $\delta y_{i\pm1}$ to be taken as the negative of the sum of the expected number of molecules moving from the $i$th compartment to the $i\pm1$ compartment and vice versa. With these, the correlation function between $\delta y_i(t)$ and $\delta y_j(t)$ can be written as

$$\langle \delta y_j(t)\delta y_i(t) \rangle = (y_{i-1}(t) + y_{i+1}(t) + 2y_i(t))d\delta t\delta_{i,j} - (y_i(t) + y_{i-1}(t))d\delta t\delta_{i-1,j} - (y_i(t) + y_{i+1}(t))d\delta t\delta_{i+1,j}. \tag{32}$$

We now take the system to continuous space by letting $y_i(t) \to \ell c(x,t)$ and $\delta_{i,j} \to \ell\delta(x-x')$ with any intances of $\pm1$ in the indices also being converted to $\pm\ell$. Putting these substitutions into *Equation 25* and dividing by $(\ell\delta t)^2$ yields

$$\begin{aligned} \left\langle \frac{\delta c(x',t)}{\delta t}\frac{\delta c(x,t)}{\delta t} \right\rangle &= \frac{d}{\delta t}((c(x-\ell,t) + c(x+\ell,t) + 2c(x,t))\delta(x-x') \\ &\quad -(c(x,t)+c(x-\ell,t))\delta(x-\ell-x') - (c(x,t)+c(x+\ell,t))\delta(x+\ell-x')) \\ &= \frac{d}{\delta t}((c(x+\ell,t)\delta(x-x') - c(x+\ell,t)\delta(x+\ell-x')) \\ &\quad -(c(x,t)\delta(x-\ell-x') - c(x,t)\delta(x-x'))(c(x,t)\delta(x-x') \\ &\quad -c(x,t)\delta(x+\ell-x')) - (c(x-\ell,t)\delta(x-\ell-x') - c(x-\ell,t)\delta(x-x'))). \end{aligned} \tag{33}$$

*Equation 33* has been rearranged into this form so as to easily apply the operators $\partial_x^\pm$ defined as

$$\partial_x^+ f(x) = \frac{f(x+\ell) - f(x)}{\ell}, \tag{34a}$$

$$\partial_x^- f(x) = \frac{f(x) - f(x-\ell)}{\ell}. \tag{34b}$$

Using this notation, *Equation 33* can be simplfied into

$$\begin{aligned} \left\langle \frac{\delta c(x',t)}{\delta t}\frac{\delta c(x,t)}{\delta t} \right\rangle &= \frac{\ell d}{\delta t}(\partial_x^+(c(x,t)\delta(x-\ell-x') - c(x,t)\delta(x-x')) \\ &\quad +\partial_x^-(c(x,t)\delta(x-x') - c(x,t)\delta(x+\ell-x'))) \\ &= \frac{\ell^2 d}{\delta t}(\partial_x^+\partial_{x'}^+ + \partial_x^-\partial_{x'}^-)(c(x,t)\delta(x-x')). \end{aligned} \tag{35}$$

Taking the $\ell \to 0$ limit while holding $D = \ell^2 d$ constant allows $\partial_x^\pm$ and $\partial_{x'}^\pm$ to converge to true derivatives, $\partial_x$ and $\partial_{x'}$. Additionally, if the $\delta c(x',t)/\delta t$ term on the left-hand side of *Equation 35* is replaced with $\delta c(x',t')/\delta t$ for $t' \neq t$, then the entire right-hand side must go to 0 as the system is Markovian. This can be accomplished by multiplying the right-hand side by a factor of $\delta_{t,t'}$. Taking the $\delta t \to 0$ limit then turns the two terms on the left-hand side into true derivatives in time, $\partial_t$ and $\partial_{t'}$, acting on $c(x,t)$ and $c(x',t')$ respectively while the factor of $\delta_{t,t'}/\delta t$ on the right-hand side becomes $\delta(t-t')$. Altogether, this transforms *Equation 35* into

$$\langle \partial_{t'}c(x',t')\partial_t c(x,t) \rangle = 2D\delta(t-t')\partial_x\partial_{x'}(c(x,t)\delta(x-x')). \tag{36}$$

Finally, by approximating the system as being in steady state, $c(x,t)$ can be replaced with $\bar{c}(x)$ and $\partial_t c(x,t)$ becomes equivalent to $\eta_D(x,t)$. Making these substitutions and generalizing *Equation 36* to arbitrary dimensions yields

$$\langle \eta_D(\vec{x}',t')\eta_D(\vec{x},t)\rangle = 2D\delta(t-t')\vec{\nabla}\cdot\vec{\nabla}'\left(\bar{c}(\vec{x})\delta^{SP}(\vec{x}-\vec{x}')\right). \tag{37}$$

Fourier transforming *Equation 37* can be easily performed due to the $\delta$ functions, integrating the spatial terms by parts, and utilizing *Equation 24* to yield

$$
\begin{aligned}
\left\langle \tilde{\eta}_D^*\left(\vec{k}',\omega'\right)\tilde{\eta}_D\left(\vec{k},\omega\right)\right\rangle &= \int d^{SP}x d^{SP}x' dt dt' e^{i\vec{k}\cdot\vec{x}}e^{-i\vec{k}'\cdot\vec{x}'}e^{i\omega t}e^{-i\omega' t'}\langle \eta_D(\vec{x}',t')\eta_D(\vec{x},t)\rangle \\
&= 2D\int d^{SP}x d^{SP}x' dt dt' e^{i\vec{k}\cdot\vec{x}}e^{-i\vec{k}'\cdot\vec{x}'}e^{i\omega t}e^{-i\omega' t'}\delta(t-t')\vec{\nabla}\cdot\vec{\nabla}'\left(\bar{c}(\vec{x})\delta^{SP}(\vec{x}-\vec{x}')\right) \\
&= 2D(2\pi\delta(\omega-\omega'))\int d^{SP}x d^{SP}x' e^{i\vec{k}\cdot\vec{x}}e^{-i\vec{k}'\cdot\vec{x}'}\vec{\nabla}\cdot\vec{\nabla}'\left(\bar{c}(\vec{x})\delta^{SP}(\vec{x}-\vec{x}')\right) \\
&= 2D(2\pi\delta(\omega-\omega'))\int d^{SP}x d^{SP}x' \bar{c}(\vec{x})\delta^{SP}(\vec{x}-\vec{x}')\vec{\nabla}\cdot\vec{\nabla}'\left(e^{i\vec{k}\cdot\vec{x}}e^{-i\vec{k}'\cdot\vec{x}'}\right) \\
&= 2D\vec{k}\cdot\vec{k}'(2\pi\delta(\omega-\omega'))\int d^{SP}x d^{SP}x' \bar{c}(\vec{x})\delta^{SP}(\vec{x}-\vec{x}')e^{i\vec{k}\cdot\vec{x}}e^{-i\vec{k}'\cdot\vec{x}'} \\
&= 2D\vec{k}\cdot\vec{k}'(2\pi\delta(\omega-\omega'))\int d^{SP}x\,\bar{c}(\vec{x})e^{i\vec{x}\cdot(\vec{k}-\vec{k}')} \\
&= 2D\vec{k}\cdot\vec{k}'\tilde{\bar{c}}\left(\vec{k}-\vec{k}'\right)(2\pi\delta(\omega-\omega')) \\
&= \frac{2\lambda^2\vec{k}\cdot\vec{k}'}{1+\lambda^2\left|\vec{k}-\vec{k}'\right|^2}\left(\beta(2\pi)^{SO+1}\delta(\omega-\omega')\delta^{SO}\left(\vec{k}-\vec{k}'\right)\right).
\end{aligned} \tag{38}
$$

Moving on to $\eta_\nu$, its correlation function must be $\delta$ correlated in time and space since it is a purely local reaction and as such, at steady state, must take the form

$$\langle \eta_\nu(\vec{x}',t')\eta_\nu(\vec{x},t)\rangle = \nu\bar{c}(\vec{x})\delta(t-t')\delta^{SP}(\vec{x}-\vec{x}'). \tag{39}$$

Fourier transforming *Equation 39* is again easily performed due to the $\delta$ functions and *Equation 24*. This yields

$$
\begin{aligned}
\left\langle \tilde{\eta}_\nu^*\left(\vec{k}',\omega'\right)\tilde{\eta}_\nu\left(\vec{k},\omega\right)\right\rangle &= \int d^{SP}x d^{SP}x' dt dt' e^{i\vec{k}\cdot\vec{x}}e^{-i\vec{k}'\cdot\vec{x}'}e^{i\omega t}e^{-i\omega' t'}\langle \eta_\nu(\vec{x}',t')\eta_\nu(\vec{x},t)\rangle \\
&= \nu\int d^{SP}x d^{SP}x' dt dt' e^{i\vec{k}\cdot\vec{x}}e^{-i\vec{k}'\cdot\vec{x}'}e^{i\omega t}e^{-i\omega' t'}\bar{c}(\vec{x})\delta(t-t')\delta^{SP}(\vec{x}-\vec{x}') \\
&= \nu(2\pi\delta(\omega-\omega'))\int d^{SP}x d^{SP}x' e^{i\vec{k}\cdot\vec{x}}e^{-i\vec{k}'\cdot\vec{x}'}\bar{c}(\vec{x})\delta^{SP}(\vec{x}-\vec{x}') \\
&= \nu(2\pi\delta(\omega-\omega'))\int d^{SP}x\,e^{i\vec{x}\cdot(\vec{k}-\vec{k}')}\bar{c}(\vec{x}) \\
&= \nu\tilde{\bar{c}}\left(\vec{k}-\vec{k}'\right)(2\pi\delta(\omega-\omega')) \\
&= \frac{1}{1+\lambda^2\left|\vec{k}-\vec{k}'\right|^2}\left(\beta(2\pi)^{SO+1}\delta(\omega-\omega')\delta^{SO}\left(\vec{k}-\vec{k}'\right)\right).
\end{aligned} \tag{40}
$$

Finally, the cross spectrum of $\eta_\beta$ must be $\delta$ correlated in $\omega$-space as well as all source dimensions of $k$-space since it is merely a uniform production term that does not depend on space or time. This yields

$$\left\langle \tilde{\eta}_\beta^*\left(\vec{k}',\omega'\right)\tilde{\eta}_\beta\left(\vec{k},\omega\right)\right\rangle = \beta(2\pi)^{SO+1}\delta(\omega-\omega')\delta^{SO}\left(\vec{k}-\vec{k}'\right). \tag{41}$$

Combining *Equations 29, 38, 40, and 41* then yields

$$\left\langle \tilde{\delta c}^*\left(\vec{k}',\omega'\right)\tilde{\delta c}\left(\vec{k},\omega\right)\right\rangle = \frac{\beta(2\pi)^{SO+1}\delta(\omega-\omega')\delta^{SO}\left(\vec{k}-\vec{k}'\right)}{\nu^2\left(1+\lambda^2\left|\vec{k}\right|^2-i\frac{\omega}{\nu}\right)\left(1+\lambda^2\left|\vec{k}'\right|^2+i\frac{\omega'}{\nu}\right)}$$

$$\cdot\left(\frac{2\lambda^2\vec{k}\cdot\vec{k}'}{1+\lambda^2\left|\vec{k}-\vec{k}'\right|^2}+\frac{1}{1+\lambda^2\left|\vec{k}-\vec{k}'\right|^2}+1\right)$$

$$=\frac{\beta(2\pi)^{SO+1}\delta(\omega-\omega')\delta^{SO}\left(\vec{k}-\vec{k}'\right)}{\nu^2\left(1+\lambda^2\left|\vec{k}\right|^2-i\frac{\omega}{\nu}\right)\left(1+\lambda^2\left|\vec{k}'\right|^2+i\frac{\omega'}{\nu}\right)}\frac{2+\lambda^2\left(\left|\vec{k}\right|^2+\left|\vec{k}'\right|^2\right)}{1+\lambda^2\left|\vec{k}-\vec{k}'\right|^2}$$

$$=\frac{\tilde{\bar{c}}\left(\vec{k}-\vec{k}'\right)\left(2\pi\delta(\omega-\omega')\right)\left(2+\lambda^2\left(\left|\vec{k}\right|^2+\left|\vec{k}'\right|^2\right)\right)}{\nu\left(1+\lambda^2\left|\vec{k}\right|^2-i\frac{\omega}{\nu}\right)\left(1+\lambda^2\left|\vec{k}'\right|^2+i\frac{\omega'}{\nu}\right)}. \tag{42}$$

We now define $m$ as

$$m(\vec{x},t)=\int_{V(a)}d^{SP}r\, c(\vec{x}+\vec{r},t), \tag{43}$$

where $V(a)$ is a $SP$-dimensional sphere with radius $a$. This allows the mean value of $m$ to be written as

$$\begin{aligned}\bar{m}(\vec{x}) &= \int_{V(a)}d^{SP}r\,\bar{c}(\vec{x}+\vec{r})=\frac{\beta\lambda^{2-(SP-SO)}}{D}\int_{V(a)}d^{SP}r\,P_{SP-SO}\left(\frac{|\vec{x}+\vec{r}|}{\lambda}\right)\\ &=\frac{\beta\lambda^{SO}}{\nu}M_{SP-SO,SP}\left(\frac{|\vec{x}|}{\lambda},\frac{a}{\lambda}\right),\end{aligned} \tag{44}$$

where

$$M_{N,N'}(x,y)=\int_{V(y)}d^{N'}u\,P_N(|\vec{x}+\vec{u}|). \tag{45}$$

Since $P_N(|\vec{x}|)$ can only depend on the last $N$ dimensions of its input vectors, the same must be true of $M_{N,N'}$. From here we define $S(\vec{x})$ as the 0-frequency limit of the cross spectrum in $\omega$-space of $m$. This allows the time averaged variance, $\sigma^2(x)$ to take the form

$$
\begin{aligned}
\sigma^2(x) \;&= \frac{S(\vec{x})}{T} = \frac{1}{T}\lim_{\omega\to 0}\int \frac{d\omega'}{2\pi}\left\langle \widetilde{\delta m}^*(\vec{x},\omega')\widetilde{\delta m}(\vec{x},\omega)\right\rangle \\[4pt]
&= \frac{1}{T}\lim_{\omega\to 0}\int \frac{d\omega'}{2\pi}\int_{V(a)} d^{SP}r\, d^{SP}r' \int \frac{d^{SP}k}{(2\pi)^{SP}}\frac{d^{SP}k'}{(2\pi)^{SP}} e^{-i\vec{k}\cdot(\vec{x}+\vec{r})}e^{i\vec{k}'\cdot(\vec{x}+\vec{r}')}\left\langle \widetilde{\delta c}^*\!\left(\vec{k}',\omega'\right)\widetilde{\delta c}\!\left(\vec{k},\omega\right)\right\rangle \\[4pt]
&= \frac{1}{(2\pi)^{2SP}\nu T}\int_{V(a)} d^{SP}r\, d^{SP}r' \int d^{SP}k\, d^{SP}k'\, e^{-i\vec{k}\cdot(\vec{x}+\vec{r})}e^{i\vec{k}'\cdot(\vec{x}+\vec{r}')} \\[4pt]
&\quad \cdot \frac{\bar{\tilde{c}}\!\left(\vec{k}-\vec{k}'\right)\left(2+\lambda^2\left(\left|\vec{k}\right|^2+\left|\vec{k}'\right|^2\right)\right)}{\left(1+\lambda^2\left|\vec{k}\right|^2\right)\left(1+\lambda^2\left|\vec{k}'\right|^2\right)} \\[4pt]
&= \frac{1}{(2\pi)^{2SP}\nu T}\int_{V(a)} d^{SP}r\, d^{SP}r' \int d^{SP}k\, d^{SP}k'\, d^{SP}z\, e^{-i\vec{k}\cdot(\vec{x}+\vec{r})}e^{i\vec{k}'\cdot(\vec{x}+\vec{r}')}e^{i\vec{z}\cdot\left(\vec{k}-\vec{k}'\right)} \\[4pt]
&\quad \cdot\bar{c}(\vec{z})\frac{2+\lambda^2\left(\left|\vec{k}\right|^2+\left|\vec{k}'\right|^2\right)}{\left(1+\lambda^2\left|\vec{k}\right|^2\right)\left(1+\lambda^2\left|\vec{k}'\right|^2\right)} \\[4pt]
&= \frac{1}{(2\pi)^{2SP}\nu T}\int_{V(a)} d^{SP}r\, d^{SP}r' \int d^{SP}k\, d^{SP}k'\, d^{SP}z\, e^{-i\vec{k}\cdot(\vec{x}+\vec{r}-\vec{z})}e^{i\vec{k}'\cdot(\vec{x}+\vec{r}'-\vec{z})} \\[4pt]
&\quad \cdot\bar{c}(\vec{z})\left(\frac{1}{1+\lambda^2\left|\vec{k}\right|^2}+\frac{1}{1+\lambda^2\left|\vec{k}'\right|^2}\right) \\[4pt]
&= \frac{1}{(2\pi)^{2SP}\nu T}\int_{V(a)} d^{SP}r\, d^{SP}r' \int d^{SP}z\, \bar{c}(\vec{z})\left(\int d^{SP}k\, e^{-i\vec{k}\cdot(\vec{x}+\vec{r}-\vec{z})}\frac{(2\pi)^{SP}\delta^{SP}(\vec{x}+\vec{r}'-\vec{z})}{1+\lambda^2\left|\vec{k}\right|^2}\right. \\[4pt]
&\quad \left. + \int d^{SP}k'\, e^{i\vec{k}'\cdot(\vec{x}+\vec{r}'-\vec{z})}\frac{(2\pi)^{SP}\delta^{SP}(\vec{x}+\vec{r}-\vec{z})}{1+\lambda^2\left|\vec{k}'\right|^2}\right) \\[4pt]
&= \frac{\beta\lambda^{2-(SP-SO)}}{D\nu\lambda^{SP}T}\int_{V(a)} d^{SP}r\, d^{SP}r' \int d^{SP}z\, P_{SP-SO}\!\left(\frac{|\vec{z}|}{\lambda}\right) \\[4pt]
&\quad \cdot\left(\delta^{SP}(\vec{x}+\vec{r}'-\vec{z})P_{SP}\!\left(\frac{|\vec{x}+\vec{r}-\vec{z}|}{\lambda}\right)+\delta^{SP}(\vec{x}+\vec{r}-\vec{z})P_{SP}\!\left(\frac{|\vec{x}+\vec{r}'-\vec{z}|}{\lambda}\right)\right) \\[4pt]
&= \frac{\beta\lambda^{4-(2SP-SO)}}{D^2T}\int_{V(a)} d^{SP}r\, d^{SP}r'\, P_{SP}\!\left(\frac{|\vec{r}-\vec{r}'|}{\lambda}\right)\left(P_{SP-SO}\!\left(\frac{|\vec{x}+\vec{r}|}{\lambda}\right)+P_{SP-SO}\!\left(\frac{|\vec{x}+\vec{r}'|}{\lambda}\right)\right) \\[4pt]
&= \frac{\beta\lambda^{4-(SP-SO)}}{D^2T}\left(\int_{V(a)} d^{SP}r\, M_{SP,SP}\!\left(\frac{|\vec{r}|}{\lambda},\frac{a}{\lambda}\right)P_{SP-SO}\!\left(\frac{|\vec{x}+\vec{r}|}{\lambda}\right)\right. \\[4pt]
&\quad \left. + \int_{V(a)} d^{SP}r'\, M_{SP,SP}\!\left(\frac{|\vec{r}'|}{\lambda},\frac{a}{\lambda}\right)P_{SP-SO}\!\left(\frac{|\vec{x}+\vec{r}'|}{\lambda}\right)\right) \\[4pt]
&= \frac{2\beta\lambda^{4-(SP-SO)}}{D^2T}\int_{V(a)} d^{SP}r\, M_{SP,SP}\!\left(\frac{|\vec{r}|}{\lambda},\frac{a}{\lambda}\right)P_{SP-SO}\!\left(\frac{|\vec{x}+\vec{r}|}{\lambda}\right) \\[4pt]
&= \frac{2\beta\lambda^{SO}}{\nu^2T}\Sigma_{SP-SO,SP}\!\left(\frac{|\vec{x}|}{\lambda},\frac{a}{\lambda}\right) = \frac{2\bar{m}(\vec{x})}{\nu T}\frac{\Sigma_{SP-SO,SP}\!\left(\frac{|\vec{x}|}{\lambda},\frac{a}{\lambda}\right)}{M_{SP-SO,SP}\!\left(\frac{|\vec{x}|}{\lambda},\frac{a}{\lambda}\right)},
\end{aligned}
\tag{46}
$$

where

$$
\Sigma_{N,N'}(x,y) = \int_{V(y)} d^{N'}u\, M_{N',N'}(u,y)P_N(|\vec{x}+\vec{u}|).
\tag{47}
$$

Wherein once again only the last *N* dimensions of the input vectors can be taken into account. Combining *Equations 44* and *46* yields the full precision to be

$$P^2(\vec{x}) = \frac{\bar{m}^2(\vec{x})}{\sigma^2(\vec{x})}\left(\frac{\Delta\bar{m}(\vec{x})}{\bar{m}(\vec{x})}\right)^2 = \frac{T}{2\tau}\frac{\beta}{\nu}M_{SP-SO,SP}\left(\frac{|\vec{x}|}{\lambda},\frac{a}{\lambda}\right)\left(1 - \frac{M_{SP-SO,SP}\left(\frac{|\vec{x}|+2a}{\lambda},\frac{a}{\lambda}\right)}{M_{SP-SO,SP}\left(\frac{|\vec{x}|}{\lambda},\frac{a}{\lambda}\right)}\right)^2, \tag{48}$$

where

$$\tau = \frac{1}{\nu}\frac{\Sigma_{SP-SO,SP}\left(\frac{|\vec{x}|}{\lambda},\frac{a}{\lambda}\right)}{M_{SP-SO,SP}\left(\frac{|\vec{x}|}{\lambda},\frac{a}{\lambda}\right)}. \tag{49}$$

With *Equation 48*, once the forms of $P_N$, $M_{N,N'}$, and $\Sigma_{N,N'}$ are determined for a given *SP* and *SO*, the full form of the noise-to-signal ratio can be found. We now calculate these forms for specific choices of *SP* and *SO*.

## 1D space, 0D source

To begin, we start with the simple scenario in which $SP = 1$ and $SO = 0$. This allows $P_1$, $M_{1,1}$, and $\Sigma_{1,1}$ to take the forms

$$P_1(x) = \int\frac{du}{2\pi}e^{-iux}\frac{1}{1+u^2} = \frac{1}{2}e^{\%}-|x| \tag{50}$$

$$\begin{aligned}M_{1,1}(x,y) &= \int_{-y}^{y}du P_1(|x+u|) = \frac{1}{2}\int_{-y}^{y}due^{-|x+u|} \\ &= \begin{cases} 1 - e^{-y}\cosh(x) & x<y \\ e^{-x}\sinh(y) & x \geq y \end{cases}\end{aligned} \tag{51}$$

$$\begin{aligned}\Sigma_{1,1}(x,y) &= \int_{-y}^{y}du M_{1,1}(u,y)P_1(|x+u|) = \frac{1}{2}\int_{-y}^{y}du(1-e^{-y}\cosh(u))e^{-|x+u|} \\ &= \begin{cases} 1 - \frac{1}{4}e^{-y}\left((5+2y-e^{-2y})\cosh(x) - 2x\sinh(x)\right) & x<y \\ \frac{1}{4}e^{-x}(4\sinh(y) - e^{-y}(2y+\sinh(2y))) & x \geq y \end{cases}\end{aligned} \tag{52}$$

*Equation 51* and *Equation 52* can then be put into *Equation 48* along with the assumption $|x|>a$ to obtain

$$P^2(x) = \frac{\bar{m}(x)T}{2\tau}\left(1 - e^{-\frac{2a}{\lambda}}\right)^2, \tag{53}$$

and

$$\tau = \frac{1}{\nu}\left(1 - e^{-\frac{a}{\lambda}}\frac{\frac{2a}{\lambda}+\sinh\left(\frac{2a}{\lambda}\right)}{4\sinh\left(\frac{a}{\lambda}\right)}\right) \tag{54}$$

as in *Equations 9 and 10* of the main text.

Next we apply the condition given by *Equation 17* to determine the regime in which these results are valid for the $SP = 1$ and $SO = 0$ case. A similar methodology can be done for each of the other cases we will look at, though this is the only one we do explicitly. To begin, we will reperform the calculation done in *Equation 46* but without taking the $\omega \to 0$ limit so as to obtain the full form of $S(\omega, x)$.

$$
\begin{aligned}
S(\omega,x) \; &= \int \frac{d\omega'}{2\pi} \langle \delta \tilde{m}^*(x,\omega') \delta \tilde{m}(x,\omega) \rangle \\
&= \int \frac{d\omega'}{2\pi} \int_{-a}^{a} dr dr' \int \frac{dk\, dk'}{2\pi\, 2\pi} e^{-ik(x+r)} e^{ik'(x+r')} \langle \delta \tilde{c}^*(k',\omega') \delta \tilde{c}(k,\omega) \rangle \\
&= \frac{1}{(2\pi)^2 \nu} \int_{-a}^{a} dr dr' \int dk dk' e^{-ik(x+r)} e^{ik'(x+r')} \frac{\tilde{\bar{c}}(k-k')\left(2+\lambda^2\left(k^2+k'^2\right)\right)}{\left(1+\lambda^2 k^2 - i\frac{\omega}{\nu}\right)\left(1+\lambda^2 k'^2 + i\frac{\omega}{\nu}\right)} \\
&= \frac{1}{(2\pi)^2 \nu} \int_{-a}^{a} dr dr' \int dk dk' dz\, e^{-ik(x+r)} e^{ik'(x+r')} e^{iz(k-k')} \bar{c}(z) \frac{2+\lambda^2\left(k^2+k'^2\right)}{\left(1+\lambda^2 k^2 - i\frac{\omega}{\nu}\right)\left(1+\lambda^2 k'^2 + i\frac{\omega}{\nu}\right)} \\
&= \frac{1}{(2\pi)^2 \nu} \int_{-a}^{a} dr dr' \int dk dk' dz\, e^{-ik(x+r-z)} e^{ik'(x+r'-z)} \bar{c}(z) \left( \frac{1}{1+\lambda^2 k^2 - i\frac{\omega}{\nu}} + \frac{1}{1+\lambda^2 k'^2 + i\frac{\omega}{\nu}} \right) \\
&= \frac{1}{(2\pi)^2 \nu} \int_{-a}^{a} dr dr' \int dz\, \bar{c}(z) \left( \int dk\, e^{-ik(x+r-z)} \frac{2\pi\delta(x+r'-z)}{1+\lambda^2 k^2 - i\frac{\omega}{\nu}} + \int dk'\, e^{ik'(x+r'-z)} \frac{2\pi\delta(x+r-z)}{1+\lambda^2 k'^2 + i\frac{\omega}{\nu}} \right) \\
&= \frac{1}{2\pi\nu} \int_{-a}^{a} dr dr' \left( \int dk\, e^{-ik(r-r')} \frac{\bar{c}(x+r')}{1+\lambda^2 k^2 - i\frac{\omega}{\nu}} + \int dk\, e^{ik(r'-r)} \frac{\bar{c}(x+r)}{1+\lambda^2 k^2 + i\frac{\omega}{\nu}} \right) \\
&= \frac{1}{\nu\lambda} \int_{-a}^{a} dr dr' \left( \bar{c}(x+r') Q\left(\frac{r-r'}{\lambda}, -\frac{\omega}{\nu}\right) + \bar{c}(x+r) Q\left(\frac{r-r'}{\lambda}, \frac{\omega}{\nu}\right) \right),
\end{aligned}
\tag{55}
$$

where

$$
Q(x,y) = \int \frac{du}{2\pi} e^{-iux} \frac{1}{1+iy+u^2} = \frac{1}{2\sqrt{1+iy}} e^{-|x|\sqrt{1+iy}}
\tag{56}
$$

and $\sqrt{1+iy}$ is assumed to be the branch with a positive real component. Plugging this and the explicit form of $\bar{c}$ into *Equation 55* then yields

$$
\begin{aligned}
S(\omega,x) \; &= \frac{\beta}{4\nu D} \int_{-a}^{a} dr dr' \left( \frac{1}{\sqrt{1-i\frac{\omega}{\nu}}} e^{-\frac{|x+r'|}{\lambda}} e^{-\frac{|r-r'|}{\lambda}\sqrt{1-i\frac{\omega}{\nu}}} + \frac{1}{\sqrt{1+i\frac{\omega}{\nu}}} e^{-\frac{|x+r|}{\lambda}} e^{-\frac{|r-r'|}{\lambda}\sqrt{1+i\frac{\omega}{\nu}}} \right) \\
&= \frac{\beta}{2\nu D} \mathrm{Re}\left[ \int_{-a}^{a} dr dr' \frac{1}{\sqrt{1+i\frac{\omega}{\nu}}} e^{-\frac{|x+r|}{\lambda}} e^{-\frac{|r-r'|}{\lambda}\sqrt{1+i\frac{\omega}{\nu}}} \right] = \frac{2\beta}{\nu^2} \mathrm{Re}\left[ \Upsilon\left(\frac{|x|}{\lambda}, \frac{a}{\lambda}, \sqrt{1+i\frac{\omega}{\nu}}\right) \right],
\end{aligned}
\tag{57}
$$

where

$$
\Upsilon(x,y,w) = \int_{-y}^{y} du\, du' \frac{1}{4w} e^{-|x+u|} e^{-w|u-u'|}.
\tag{58}
$$

The function $\Upsilon(x,y,w)$ limits to $\Sigma_{1,1}(x,y)$ when $w \to 1$ and as such has different forms when $x$ is less or greater than $y$. As the purpose of this exercise is to determine the regime in which our theoretical approximations are valid and our model obeys $|x| \geq 2a$ for all cells, here we will only present the $x>y$ solution for simplicity. Using this to perform the integrals in *Equation 58* and applying the result to *Equation 57* then yields

$$
S(\omega,x) = \frac{2\beta}{\nu^2} e^{-\frac{|x|}{\lambda}} \mathrm{Re}\left[ \frac{1}{W^2} \sinh\left(\frac{a}{\lambda}\right) - e^{-\frac{a}{\lambda}W} \frac{W \sinh\left(\frac{a}{\lambda}W\right)\cosh\left(\frac{a}{\lambda}\right) - \cosh\left(\frac{a}{\lambda}W\right)\sinh\left(\frac{a}{\lambda}\right)}{W^2(W^2-1)} \right],
\tag{59}
$$

where $W = \sqrt{1+i\frac{\omega}{\nu}}$, which in turn implies $\omega = -i\nu(W^2-1)$. With this, it is easier to perform all further calculations with respect to $W$ and take the $W \to 1$ limit as that is equivalent to the $\omega \to 0$ limit.

We can now combine this with the known form of $S(0,x)$ given in *Equation 46* to evaluate *Equation 17* to take the form

$$
\begin{aligned}
T^2 \; &\gg \left| \frac{6}{S} \frac{\partial^2 S}{\partial \omega^2} \Big|_{\omega=0} \right| = \left| \frac{6}{S} \frac{\partial W}{\partial \omega} \frac{\partial}{\partial W} \left( \frac{\partial W}{\partial \omega} \frac{\partial S}{\partial W} \right) \Big|_{W=1} \right| = \left| \frac{6}{S} \left( \left(\frac{\partial \omega}{\partial W}\right)^{-2} \frac{\partial^2 S}{\partial W^2} - \left(\frac{\partial \omega}{\partial W}\right)^{-3} \frac{\partial^2 \omega}{\partial W^2} \frac{\partial S}{\partial W} \right) \Big|_{W=1} \right| \\
&= \frac{1}{2\nu^2} \frac{96 \sinh\left(\frac{a}{\lambda}\right) - e^{-\frac{a}{\lambda}}\left(48\frac{a}{\lambda} + 30\left(\frac{a}{\lambda}\right)^2 + 8\left(\frac{a}{\lambda}\right)^3 + 33\sinh\left(2\frac{a}{\lambda}\right)\right) + 6e^{-3\frac{a}{\lambda}}\left(3\frac{a}{\lambda} + \left(\frac{a}{\lambda}\right)^2\right)}{4\sinh\left(\frac{a}{\lambda}\right) - e^{-\frac{a}{\lambda}}\left(2\frac{a}{\lambda} + \sinh\left(2\frac{a}{\lambda}\right)\right)}.
\end{aligned}
\tag{60}
$$

The right-hand side of **Equation 60** is a function that monotonically increases from $9/2\nu^2$ to $21/2\nu^2$ as $a/\lambda$ goes from 0 to $\infty$. Thus, regardless of the value of $\lambda$, $\nu$ sets the timescale to which $T$ must be compared.

## 2D space, 0D source

For $SP = 2$ and $SO = 0$, $P_2$, $M_{2,2}$, and $\Sigma_{2,2}$ each take the form

$$P_2(x) = \int \frac{d^2 u}{(2\pi)^2} e^{-i\vec{u}\cdot\vec{x}} \frac{1}{1+|\vec{u}|^2} = \frac{1}{2\pi} K_0(x) \tag{61}$$

$$
\begin{aligned}
M_{2,2}(x,y) &= \int_{V(y)} d^2 u P_2(|\vec{x}+\vec{u}|) = \int_{V(y)} d^2 u \int \frac{d^2 u'}{(2\pi)^2} e^{-i\vec{u}'\cdot(\vec{x}+\vec{u})} \frac{1}{1+|\vec{u}'|^2} \\
&= y \int_0^\infty du' \frac{J_0(xu')J_1(yu')}{1+u'^2}
\end{aligned}
\tag{62}
$$

$$
\begin{aligned}
\Sigma_{2,2}(x,y) &= \int_{V(y)} d^2 u M_{2,2}(|\vec{u}|,y) P_2(|\vec{x}+\vec{u}|) \\
&= y \int_{V(y)} d^2 u \int_0^\infty du' \int \frac{d^2 u''}{(2\pi)^2} \frac{J_0(|\vec{u}|u')J_1(yu')}{1+u'^2} \frac{e^{-i\vec{u}''\cdot(\vec{x}+\vec{u})}}{1+|\vec{u}''|^2} \\
&= y^2 \int_0^\infty du'\, du'' \frac{u'' J_0(xu'')J_1(yu')(u'J_0(yu'')J_1(yu') - u''J_0(yu')J_1(yu''))}{(u'^2-u''^2)(1+u'^2)(1+u''^2)},
\end{aligned}
\tag{63}
$$

where $J_n(x)$ and $K_n(x)$ are the Bessel functions of the first kind and modified Bessel functions of the second kind, respectively. Unfortunately, the complicated nature of Bessel functions makes the remaining integrals unsolvable analytically, and therefore we evaluate them numerically. Similar problems arise whenever $SP = 2$ or $SP - SO = 2$.

## 3D space, 0D source

For $SP = 3$ and $SO = 0$, $P_3$, $M_{3,3}$, and $\Sigma_{3,3}$ each take the form

$$P_3(x) = \int \frac{d^3 u}{(2\pi)^3} e^{-i\vec{u}\cdot\vec{x}} \frac{1}{1+|\vec{u}|^2} = \frac{1}{4\pi x} e^{-x} \tag{64}$$

$$
\begin{aligned}
M_{3,3}(x,y) &= \int_{V(y)} d^3 u P_3(|\vec{x}+\vec{u}|) = \frac{1}{4\pi} \int_{V(y)} d^3 u \frac{1}{|\vec{x}+\vec{u}|} e^{-|\vec{x}+\vec{u}|} \\
&= \begin{cases} 1 - \frac{1+y}{x} e^{-y}\sinh(x) & x < y \\ \frac{1}{x} e^{-x}(y\cosh(y) - \sinh(y)) & x \geq y \end{cases}
\end{aligned}
\tag{65}
$$

$$
\begin{aligned}
\Sigma_{3,3}(x,y) &= \int_{V(y)} d^3 u M_{3,3}(|\vec{u}|,y) P_3(|\vec{x}+\vec{u}|) \\
&= \frac{1}{4\pi} \int_{V(y)} d^3 u \left(1 - \frac{1+y}{|\vec{u}|} e^{-y}\sinh(|\vec{u}|)\right) \frac{1}{|\vec{x}+\vec{u}|} e^{-|\vec{x}+\vec{u}|} \\
&= \begin{cases} 1 - \frac{1}{4x} e^{-y}(1+y)\left((5+2y+e^{-2y})\sinh(x) - 2x\cosh(x)\right) & x < y \\ \frac{1}{4x} e^{-x}(4(y\cosh(y) - \sinh(y)) + e^{-y}(1+y)(2y - \sinh(2y))) & x \geq y \end{cases}
\end{aligned}
\tag{66}
$$

## 2D space, 1D source

For $SP = 2$, $SO = 1$, $P_1$ and $M_{2,2}$ are known from **Equations 50** and **Equations 62**. This leaves $M_{1,2}$ and $\Sigma_{1,2}$ to take the forms

$$M_{1,2}(x,y) = \int_{V(y)} d^2u P_1(|\vec{x}+\vec{u}|) = \frac{1}{2}\int_0^y du \int_0^{2\pi} d\theta u e^{-|x_2+u_2|}$$

$$= e^{-|x_2|}\int_0^{2\pi} d\theta \frac{1-e^{-y\sin(\theta)}(1+y\sin(\theta))}{2(\sin(\theta))^2} \tag{67}$$

$$\Sigma_{1,2}(x,y) = \int_{V(y)} d^2u M_{2,2}(|\vec{u}|,y) P_1(|\vec{x}+\vec{u}|)$$

$$= \frac{y}{2}\int_0^y du \int_0^{2\pi} d\theta \int_0^\infty du' u \frac{J_0(uu')J_1(yu')}{1+u'^2} e^{-|x_2+u\sin(\theta)|} \tag{68}$$

Again, we evaluate the remaining integrals numerically.

## 3D space, 2D source

For $SP=3$, $SO=2$, $P_1$ and $M_{3,3}$ are known from **Equations 50** and **Equations 65**. This leaves $M_{1,3}$ and $\Sigma_{1,3}$ to take the forms

$$M_{1,3}(x,y) = \int_{V(y)} d^3u P_1(|\vec{x}+\vec{u}|) = \frac{1}{2}\int_{V(y)} d^3u e^{-|x_3+u_3|}$$

$$= 2\pi \begin{cases} e^{-y}(1+y)\cosh(x) + \dfrac{y^2-x^2}{2} - 1 & x<y \\ e^{-x}(y\cosh(y)-\sinh(y)) & x \geq y \end{cases} \tag{69}$$

$$\Sigma_{1,3}(x,y) = \int_{V(y)} d^3u M_{3,3}(|\vec{u}|,y) P_1(|\vec{x}+\vec{u}|) = \frac{1}{2}\int_{V(y)} d^3u \left(1 - \frac{1+y}{|\vec{u}|}e^{-y}\sinh(|\vec{u}|)\right) e^{-|x_3+u_3|}$$

$$= 2\pi \begin{cases} e^{-y}(1+y)\left(\dfrac{7+2y+e^{-2y}}{4}\cosh(x) - \dfrac{x}{2}\sinh(x) - \cosh(y)\right) + \dfrac{y^2-x^2}{2} - 1 & x<y \\ e^{-x}\left(\dfrac{4y^2+5y-1}{8}e^{-y} + \dfrac{1+y}{8}e^{-3y} + \dfrac{3y}{4}\cosh(y) - \dfrac{5}{4}\sinh(y)\right) & x \geq y \end{cases} \tag{70}$$

## Appendix 2

### Hopping model for SDC case

To obtain a more intuitive understanding of why the SDC model results in the scaling properties seen in the various calculations of $M_{\mathrm{SP-SO,SP}}$ and $\Sigma_{\mathrm{SP-SO,SP}}$, we now look at a simpler version of one dimensional diffusion in which we discretize space into compartments of uniform size. Let molecules still be produced in the 0th compartment at rate $\beta$ and degrade anywhere in space at rate $\nu$. The process of diffusion can be approximated by letting the molecules hop to neighboring compartments with rate $h$ with equal probability of moving left or right. This allows the dynamics of $m_j$, the number of molecules in the $j$ compartment for $j \in \mathbb{Z}$, to be written as

$$\frac{\partial m_j}{\partial t} = \beta \delta_{0j} + h\left(m_{j+1} + m_{j-1} - 2m_j\right) - \nu m_j. \tag{71}$$

By setting the left-hand side of *Equation 45* to 0, the resulting system of equations can be easily solved by assuming $\bar{m}_j = A\exp(-2|j|/\lambda)$ and calculating $A$ and $\lambda$. Imposing this assumption on *Equation 45* and taking $j>0$ yields

$$\begin{aligned}
0 &= h\left(Ae^{-\frac{2(j+1)}{\lambda}} + Ae^{-\frac{2(j-1)}{\lambda}} - 2Ae^{-\frac{2j}{\lambda}}\right) - \nu Ae^{-\frac{2j}{\lambda}} = Ae^{-\frac{2j}{\lambda}}\left(he^{-\frac{2}{\lambda}} + he^{\frac{2}{\lambda}} - 2h - \nu\right) \\
&= Ae^{-\frac{2j}{\lambda}}\left(4h\sinh^2\left(\frac{1}{\lambda}\right) - \nu\right) \Longrightarrow \lambda = \mathrm{asinh}^{-1}\left(\sqrt{\frac{\nu}{4h}}\right).
\end{aligned} \tag{72}$$

With $\lambda$ solved for, we solve for the proportionality constant by noting that the total number of molecules in the whole system must follow a simple birth-death process with a mean of $\beta/\nu$. This in turn implies

$$\begin{aligned}
\frac{\beta}{\nu} &= \sum_{j=-\infty}^{\infty} Ae^{-\frac{2|j|}{\lambda}} = A\left(2\left(\sum_{j=0}^{\infty} e^{-\frac{2j}{\lambda}}\right) - 1\right) = A\left(\frac{2}{1-e^{-\frac{2}{\lambda}}} - 1\right) = A\left(\frac{e^{\frac{1}{\lambda}}}{\sinh\left(\frac{1}{\lambda}\right)} - 1\right) = A\coth\left(\frac{1}{\lambda}\right) \\
&\Longrightarrow A = \frac{\beta}{\nu}\tanh\left(\frac{1}{\lambda}\right),
\end{aligned} \tag{73}$$

This in turn gives the average value of $m_j$ to be

$$\bar{m}_j = \frac{\beta}{\nu}\tanh\left(\frac{1}{\lambda}\right)e^{-\frac{2|j|}{\lambda}}. \tag{74}$$

Next, we calculate the full distribution of $m_j$ by assuming that at any given moment in time each molecule in the system has probability $P_j$ of being in the $j$th compartment. This can be combined with the aforementioned fact that $N$, the total number of molecules in the system, must follow a birth-death process and thus to Poissonianly distributed with mean $\beta/\nu$. For any given value of $N$, $P(m_j|N)$ must be a binomial distribution with success probability $P_j$ since each molecule is independent. This allows the marginal distribution $P(m_j)$ to be calculated to be

$$\begin{aligned}
P(m_j) &= \sum_{N=m_j}^{\infty} P(N)P(m_j|N) = \sum_{N=m_j}^{\infty} e^{-\frac{\beta}{\nu}}\frac{\left(\frac{\beta}{\nu}\right)^N}{N!}\binom{N}{m_j}P_j^{m_j}\left(1-P_j\right)^{N-m_j} \\
&= e^{-\frac{\beta}{\nu}}\frac{\left(\frac{\beta}{\nu}P_j\right)^{m_j}}{m_j!}\sum_{N=m_j}^{\infty}\frac{\left(\frac{\beta}{\nu}\left(1-P_j\right)\right)^{N-m_j}}{(N-m_j)!} = e^{-\frac{\beta}{\nu}}\frac{\left(\frac{\beta}{\nu}P_j\right)^{m_j}}{m_j!}e^{\frac{\beta}{\nu}\left(1-P_j\right)} = e^{-\frac{\beta}{\nu}P_j}\frac{\left(\frac{\beta}{\nu}P_j\right)^{m_j}}{m_j!}.
\end{aligned} \tag{75}$$

Thus, $m_j$ is seen to be Poissonianly distributed with mean $\beta P_j/\nu$. Comparing this mean to that derived in *Equation 46* then implies

$$P_j = \tanh\left(\frac{1}{\lambda}\right)e^{-\frac{2|j|}{\lambda}}. \tag{76}$$

We now consider the joint distribution of $m_j$ and $m_k$ for $j \neq k$. Since molecules cannot be in the $j$th and $k$th compartment simultaneously, the joint conditional distribution $P(m_j, m_k|N)$ must be trinomially distributed. This allows for the joint distribution to be calculated in a manner similar to *Equation 75* to produce

$$\begin{aligned}
P(m_j, m_k) &= \sum_{N=m_j+m_k}^{\infty} P(N)P(m_j, m_k|N) \\
&= \sum_{N=m_j+m_k}^{\infty} e^{-\frac{\beta}{\nu}} \frac{\left(\frac{\beta}{\nu}\right)^N}{N!} \binom{N}{m_j, m_k} P_j^{m_j} P_k^{m_k} \left(1 - P_j - P_k\right)^{N-m_j-m_k} \\
&= e^{-\frac{\beta}{\nu}} \frac{\left(\frac{\beta}{\nu}P_j\right)^{m_j}}{m_j!} \frac{\left(\frac{\beta}{\nu}P_k\right)^{m_k}}{m_k!} \sum_{N=m_j+m_k}^{\infty} \frac{\left(\frac{\beta}{\nu}\left(1-P_j-P_k\right)\right)^{N-m_j-m_k}}{(N-m_j-m_k)!} \\
&= e^{-\frac{\beta}{\nu}} \frac{\left(\frac{\beta}{\nu}P_j\right)^{m_j}}{m_j!} \frac{\left(\frac{\beta}{\nu}P_k\right)^{m_k}}{m_k!} e^{\frac{\beta}{\nu}\left(1-P_j-P_k\right)} = \left(e^{-\frac{\beta}{\nu}P_j} \frac{\left(\frac{\beta}{\nu}P_j\right)^{m_j}}{m_j!}\right)\left(e^{-\frac{\beta}{\nu}P_k} \frac{\left(\frac{\beta}{\nu}P_k\right)^{m_k}}{m_k!}\right).
\end{aligned} \tag{77}$$

Thus, the joint probability distribution of $m_j$ and $m_k$ is seen to be separable into the product of the two marginal distribution, meaning that same-time, instantaneous measurements of $m_j$ an $m_k$ must be uncorrelated.

From here we can begin to calculate the full correlation function for $m_j$ and $m_k$. We start by defining $\delta m_j(t) = m_j(t) - \bar{m}_j$ and $\delta m_k(t) = m_k(t) - \bar{m}_k$. Since $\bar{m}_j$ is known to set the right-hand side of *Equation 45* to 0, the dynamics of $\delta m_j$ can be written as

$$\frac{\partial \delta m_j}{\partial t} = h\left(\delta m_{j+1} + \delta m_{j-1} - 2\delta m_j\right) - \nu \delta m_j, \tag{78}$$

with the same being true for $\delta m_k$. Additionally, we assume the system is at steady state so that all mean expressions are invariant to time translation. Given this, we can without loss of generality take the correlation function between $\delta m_j$ and $\delta m_k$ to have the form

$$C_{j,k}(t) = \langle \delta m_k(t)\delta m_j(0)\rangle, \tag{79}$$

where $t > 0$. Applying the dynamic result given in *Equation 78* then yields

$$\begin{aligned}
\frac{\partial C_{j,k}}{\partial t} &= \left\langle \frac{\partial \delta m_k(t)}{\partial t}\delta m_j(0)\right\rangle = \left\langle (h(\delta m_{k+1}(t) + \delta m_{k-1}(t) - 2\delta m_k(t)) - \nu \delta m_k(t))\delta m_j(0)\right\rangle \\
&= h\left(C_{j,k+1} + C_{j,k-1}\right) - (2h+\nu)C_{j,k}.
\end{aligned} \tag{80}$$

The final form of *Equation 80* can be split into the term $-(2h+\nu)C_{j,k}$ which implies $C_{j,k} \propto \exp(-(2h+\nu)t)$ and the term $h(C_{j,k+1} + C_{j,k-1})$ which is the recursion relation for $I_\ell(2ht)$, the modified Bessel function of the first kind, where $\ell$ is some function of $j$ and $k$. This means $C_{j,k}(t)$ can be written as

$$C_{j,k}(t) = AI_{\ell(j,k)}(2ht)e^{-(2h+\nu)t}, \tag{81}$$

for some proportionality constant $A$.

To determine the forms of $A$ and $\ell(j,k)$, we can utilize the initial condition that $m_j$ is Poissonianlly distributed and thus has a variance equal to its mean while being completely uncorrelated with $m_k$ when both are measured at the same time. This means $C_{j,k}(0)$ can be written as

$$C_{j,k}(0) = \frac{\beta}{\nu}P_j\delta_{jk}, \tag{82}$$

which in turn implies $\ell(j,j) = 0$ as $I_n(0) = \delta_{0n}$ for $n \in \mathbb{Z}$. To satisfy the recursion relation term of *Equation 80*, it must then be the case that $\ell(j, j+n) = n$. Setting $k = j+n$ thus yields $\ell(j,k) = k - j$. Since $k$ and $j$ are integers, $\ell(j,k) = j - k$ is equally valid as $I_n = I_{-n}$ again for $n \in \mathbb{Z}$. Combining these results together yields the final form of $C_{j,k}(t)$ to be

$$C_{j,k}(t) = \frac{\beta}{\nu}P_jI_{k-j}(2ht)e^{-(2h+\nu)t}. \tag{83}$$

Next, let $\tau$ be the autocorrelation time of $m_j$. This quantity is typically defined by integrating $C_{j,j}(t)/C_{j,j}(0)$ over all time. Using the known properties of modified Bessel functions, this can be solved to yield

$$\tau = \int_0^\infty dt \frac{C_{j,j}(t)}{C_{j,j}(0)} = \int_0^\infty dt > I_0(2ht) e^{-(2h+\nu)t} = \frac{1}{\sqrt{\nu(4h+\nu)}}. \tag{84}$$

If we now define $M = T/\tau$ where $M$ is the number of effectively independent measurements that can be made in a time $T$, we see that for $h \gg \nu$, $M \approx 2\sqrt{\nu h}T$. Additionally, from **Equation 72** we see that in the $h \gg \nu$ regime $\lambda \approx 2\sqrt{h/\nu}$. By equating this $\lambda$ to the nondimensionalized $\lambda_{\mathrm{SDC}}/a$ from the SDC model we see that $M \approx \lambda \nu T = (\lambda_{\mathrm{SDC}}/a)\nu T$. This is consistent with the fact that for $\lambda_{\mathrm{SDC}} \gg a$ the right-hand side of **Equation 54** becomes approximately $a/\lambda_{\mathrm{SDC}}\nu$, which allows $M = T/\tau_{SDC} \approx (\lambda_{\mathrm{SDC}}/a)\nu T$.

In the $h \ll \nu$ regime, we find $M \approx \nu T$. Once again, this consistent with **Equation 54** when $\lambda_{\mathrm{SDC}} \ll a$ as this causes the right-hand side to become approximately $\nu^{-1}$. Thus, the SDC model is seen to have a correlation time that agrees with **Equation 84** in both the large and small $h$ regime.

# Appendix 3

## Comparison to experimental data

To compare our theory to experimental data, we focus on ten of the morphogens presented in Table 1 of *Kicheva et al., 2012* and obtain data from the references therein. For Bicoid, we obtain a value of $\lambda$ of ~100 $\mu$m from the text of *Gregor et al., 2007b* with and error of $\pm 10$ $\mu$m from the finding in *Gregor et al., 2007a* that cells have a ~10% error in measuring the Bicoid gradient. We then take the *a* value of the *Drosophila* embryo cells that are subjected to the Bicoid gradient to be ~2.8 $\mu$m based on Figure 3A of *Gregor et al., 2007a*. We use the same figure to estimate the size of the whole embryo to be ~500 $\mu$m or ~90 cells. This value of *a* is also used for Dorsal as measurements of both Bicoid and Dorsal occur in the *Drosophila* embryo at nuclear cycle 14. For the value of $\lambda$ for Dorsal, we use Figure 3D from *Liberman et al., 2009* to obtain a full width at 60% max of 45 $\pm$ 10 $\mu$m. Since this represents the width of Gaussian fit on both sides of the source whereas our model uses an exponential profile, we assume the appropriate $\lambda$ value for such an exponential fit would be half this value, 22.5 $\pm$ 5 $\mu$m. *Figure 3A* from the same source also shows that the distance from the ventral midline to the dorsal midline is ~200 $\mu$m or ~35 cells.

For Dpp and Wg, *Kicheva et al., 2007* provides explicit measurements of $\lambda$ for each. These values are 20.2 $\pm$ 5.7 $\mu$m and 5.8 $\pm$ 2.04 $\mu$m respectively. For Hh, we use Figure S2C in the supplementary material of *Wartlick et al., 2011* to determine $\lambda$ to be 8 $\pm$ 3 $\mu$m. Dpp, Wg, and Hh all occur in the wing disc during the third instar of the *Drosophila* development. As such, we use a common value of *a* for all three. This value is taken to be 1.3 $\mu$m based on the area of the cells being reported as 5.5 $\pm$ 0.8 $\mu$m² in the supplementary material of *Kicheva et al., 2007* and the assumption that the cells are circular. Additionally, the scale bar for Figure 1A in *Wartlick et al., 2011* shows the maximal distance from the morphogen producing midline of the wing disc to its edge to be ~250 $\mu$m or ~100 cells.

The $\lambda$ value of Fgf8 is reported as being 197 $\pm$ 7 $\mu$m in *Yu et al., 2009*. Additionally, based off the scale bars seen in Figure 2C–E of *Yu et al., 2009*, we estimate the value of *a* for the cells to be ~10 $\mu$m. For the morphogens involved in the Nodal/Lefty system (cyclops, squint, lefty1, and lefty2), measurements of $\lambda$ for each are taken from Figure 2C–F of *Müller et al., 2012* by observing where the average of the three curves crosses the 37% of max threshold with error bars given by the width of the region in which the vertical error bars of each plot intersect this threshold line. We assume the *a* value of each morphogen in the Nodal/Lefty system to be equivalent to the *a* value of cells in the Fgf8 measurements performed in *Yu et al., 2009*. This is because the measurements made in *Müller et al., 2012* were taken during the blastula stage of the zebrafish development while measurements taken in *Yu et al., 2009* we taken in the sphere germ ring stage. These stages occur at ~2.25 and ~5.67 hpf respectively, but the blastula stage can last until ~6 hpf based on the timeline of zebrafish development presented in *Kimmel et al., 1995*. As such, since there is potential overlap in the time frame of these two stages, we assume the cells maintain a relatively fixed size and thus that the value of *a* for the Nodal/Lefty system can be taken as the same value of *a* used for Fgf8. Additionally, as seen in Figures 8F and 11B in *Kimmel et al., 1995*, these two stages also share a rougly equal overall diameter of the embryo of ~500 $\mu$m at the largest point. This creates a circumference of ~1600 $\mu$m or ~80 cells, which in turn means the morphogen must travel a maximum distance of ~40 cells away from the source.

| Morphogen | Organism | $\lambda$ (µm) | *a* (µm) | *N* |
|---|---|---|---|---|
| Bicoid | *Drosophila* | 100 ± 10 | 2.8 | 90 |
| Fgf8 | Zebrafish | 197 ± 7 | 10 | 40 |
| Lefty2 | Zebrafish | 150 ± 25 | 10 | 40 |
| Lefty1 | Zebrafish | 115 ± 20 | 10 | 40 |
| Dpp | *Drosophila* | 20.2 ± 5.7 | 1.3 | 100 |

*Continued on next page*

*continued*

| Morphogen | Organism | $\lambda$ (μm) | $a$ (μm) | $N$ |
|---|---|---|---|---|
| Dorsal | *Drosophila* | 22.5 ± 5 | 2.8 | 35 |
| Squint | Zebrafish | 65 ± 10 | 10 | 40 |
| Cyclops | Zebrafish | 30 ± 5 | 10 | 40 |
| Hh | *Drosophila* | 8 ± 3 | 1.3 | 100 |
| Wg | *Drosophila* | 5.8 ± 2.04 | 1.3 | 100 |

