## [Decision Letter]

**Acceptance summary:**

During embryonic development, morphogen gradients provide positional information to differentiating cells. Experiments have demonstrated that the precision by which these gradients are created and read out is often very high, raising the question of the fundamental limits by which nuclei and cells can obtain positional information.

Fancher and Mugler compare two mechanisms for gradient formation, direct-transport and synthesis-diffusion-clearance, and show that the former is favoured for steep gradients and the latter at shallow gradients, in good qualitative agreement with observations.

**Decision letter after peer review:**

Thank you for submitting your article "Diffusion vs. direct transport in the precision of morphogen readout" for consideration by *eLife*. Your article has been reviewed by three peer reviewers, one of whom is a member of our Board of Reviewing Editors, and the evaluation has been overseen by Marianne Bronner as the Senior Editor. The reviewers have opted to remain anonymous.

The reviewers have discussed the reviews with one another and the Reviewing Editor has drafted this decision to help you prepare a revised submission.

Summary:

The authors present theoretical results concerning two canonical, alternative mechanisms for patterning of cells by morphogen gradients. The issue is that cells know about their positions in the organism, and thus their intended cell fates, from measurements of morphogen concentrations. There will necessarily be uncertainty due noise in this concentration, and evolution has presumably favored mechanisms that reduce this noise (in particular relative to signal – namely the average difference in morphogen levels between adjacent cells). The first mechanism is direct transport (DT) of morphogen from a source cell to target cells via specific channels (e.g. cytonemes). The second mechanism is simple diffusion with degradation, which they call synthesis-diffusion-clearance (SDC). By rigorously formulating noise models for these different schemes, the authors draw several interesting conclusions. First, the transport process associated with DT does not in itself contribute to the noise. This follows from the assumption that morphogen molecules are independent so that the details of transport only yield a the net rate of arrival of molecules at the target cell. More importantly, the authors discover that there is a cross-over as a function of profile length from DT as the preferred mechanism to SDC as the preferred mechanism. The authors do an excellent job of presenting intuitive arguments for this and other results. (In this case, the essential disadvantage of the DT mechanism is that cells can only glean information from the morphogens that they absorb, whereas in the SDC mechanism cells can sense all molecules that pass by them.) The authors follow up this theoretical study with an analysis of multiple developmental morphogen systems. The results are impressively consistent with their theoretical predictions concerning which mechanism has higher signal-to-noise given the profile lengths. Overall, this is an exemplary paper: it identifies an important unrecognized question, rigorously formulates and solves theoretical models to address the question, and carefully applies the results to gain original insights into well-studied developmental systems.

Essential revisions:

1) We find the notation in the paper is unnecessarily complicated. For example, all the overbars in Equation 1 are not needed if the quantities are defined to be the mean. Otherwise, we have symbols with three labels, the bar, the j and +/-. Notation in a biology journal must be very carefully chosen for maximum simplicity and clarity. Likewise, the δ functions in these equations could more clearly be shifted to statements about boundary conditions, which would be more intuitive. Remember, very few biologists understand δ functions!

2) The argument presented in the third paragraph of the subsection “Direct Transport Model” is a key one, but we find that it could be explained more clearly. Perhaps by reference to a more detailed diagram?

3) The lack of parallel construction between the presentations of the two models is odd. The DT model is presented as a set of deterministic ODEs for mean concentrations, while the diffusive model (itself an averaged equation) has noise terms added. We would have expected the statement about the noise effect in the DT model to be deduced from some underlying master equation of which the presented model is simply the first moment(s), but that higher order moments are necessary to calculate to draw conclusions about the model.

4) Stepping back from the calculations we are struck by the following question. If, as stated: "We see that regardless of the form of *p(τ)*, the probability of a morphogen molecule entering the target cell in any given small time window δ*t* is simply *β*δ*t*. This result holds regardless of the mechanism by which morphogen molecules go

from the source cell to the target cell, as the only effect such a mechanism can have is on *p(τ)*", then why can't we substitute a diffusive mechanism and deduce the same thing?

5) The authors assume that once a morphogen molecule has reached the end of the cytoneme, it is immediately absorbed by the target cell. How realistic is this assumption and how important is it? Would relaxing it change the results qualitatively?

6) In comparing the precision of the two models, it would be good to discuss in a bit more detail how the comparison is made: which parameters are kept the same, which are optimized over, which are varied systematically, and why is this choice the most natural one? For example, the authors chose to study the precision ratio as a function of the gradient length, while optimizing over the DT shape φ. Βut we could also imagine a comparison in which the production and degradation rates are kept the same for both models (because production and degradation are energetically costly), but that over all other parameters the precision is optimized. Would that give a qualitatively different result? Why would the gradient range λ be of special importance (the comparison is made on the footing of equal gradient range)? Should not only the precision matter (such that it would be fine that the gradients in the DT and SDC model are different)? In other words, how do the maximal precisions that can be reached given reasonable constraints (such as production and degradation rates) compare, as a function of the distance to the source?

7) And in making this comparison, does the SDC model exhibit an optimal diffusion constant that maximizes the precision? Understanding this is probably important, because when, in the current comparison, the SDC model is more precise, it is because of the higher protein clearance rate due to diffusion. Yet, a higher diffusion constant also lowers the steepness of the gradient, decreasing the precision,

8) The key mechanism by which the SDC model can become superior is indeed diffusion. Could adding diffusion to the DT model improve the performance of the latter model? Is there evidence for diffusion in experimental systems that employ directed transport?

9) In DT systems, is it clear that molecule transport is independent? How could correlated transport be included in the DT model?

10) Also, in the DT model could noise be reduced by cells measuring arrivals of particles rather than averaging over internal concentration of particles? This would presumably only change the resulting expression for precision by a numerical factor, but could this factor shift the balance in favor of DT in some cases?

---

## [Author Response]

Essential revisions:1) We find the notation in the paper is unnecessarily complicated. For example, all the overbars in Equation 1 are not needed if the quantities are defined to be the mean. Otherwise, we have symbols with three labels, the bar, the j and +/-. Notation in a biology journal must be very carefully chosen for maximum simplicity and clarity. Likewise, the δ functions in these equations could more clearly be shifted to statements about boundary conditions, which would be more intuitive. Remember, very few biologists understand δ functions!

We have made the suggested revisions to the notation by removing the bar notation entirely. Instead, we express stochastic variables as time dependent and their steady state means as time independent. We have also removed the *δ* functions from the dynamic equations for the DT model (Equation 1) and expressed them instead as boundary conditions. We have left the *δ* functions in the equations for the SDC model though, as our method of incorporating Langevin noise terms requires the use of *δ* functions rather than simple boundary conditions. Finally, we have changed all + and − labels to be superscripts so as to make the notation more consistent across variables.

2) The argument presented in the third paragraph of the subsection “Direct Transport Model” is a key one, but we find that it could be explained more clearly. Perhaps by reference to a more detailed diagram?

We have added more text to better clarify the necessary assumptions needed to make this argument as well as created a new figure (Figure 2) to provide a visualization.

3) The lack of parallel construction between the presentations of the two models is odd. The DT model is presented as a set of deterministic ODEs for mean concentrations, while the diffusive model (itself an averaged equation) has noise terms added. We would have expected the statement about the noise effect in the DT model to be deduced from some underlying master equation of which the presented model is simply the first moment(s), but that higher order moments are necessary to calculate to draw conclusions about the model.

We have added a paragraph just before the DT section of our results to address this potential source of confusion. Specifically, our method in both models is to solve for certain moments of the master equation. In the DT model, we only require the first moment to fully characterize the statistical properties of the morphogen molecule count in the target cells, while in the SDC model we solve for the second moment via the known properties of the noise terms introduced into the equation for the first moment.

4) Stepping back from the calculations we are struck by the following question. If, as stated: "We see that regardless of the form of p(τ), the probability of a morphogen molecule entering the target cell in any given small time window δt is simply βδt. This result holds regardless of the mechanism by which morphogen molecules gofrom the source cell to the target cell, as the only effect such a mechanism can have is on p(τ)", then why can't we substitute a diffusive mechanism and deduce the same thing?

Our SDC model violates a key assumption of the argument depicted in the new Figure 2 by allowing molecules to leave the target cell after entering it. We have added additional text both in the description of the argument itself as well as the SDC section of our results to reinforce this.

5) The authors assume that once a morphogen molecule has reached the end of the cytoneme, it is immediately absorbed by the target cell. How realistic is this assumption and how important is it? Would relaxing it change the results qualitatively?

While it is difficult to speak to the realism of this assumption, as we are unaware of any experimental study that investigates the dynamics of molecules specifically at the ends of cytonemes, we can say that it is not particularly important. Relaxing this assumption would only alter the specifics of the transport mechanism, which as we argue with our new Figure 2, does not change the noise properties.

The molecule count in the target cells in the DT model would still behave identically to a birthdeath system. The only way relaxing this assumption would qualitatively change our results would be if it caused a significant alteration to the mean profile.

6) In comparing the precision of the two models, it would be good to discuss in a bit more detail how the comparison is made: which parameters are kept the same, which are optimized over, which are varied systematically, and why is this choice the most natural one? For example, the authors chose to study the precision ratio as a function of the gradient length, while optimizing over the DT shape φ. Βut we could also imagine a comparison in which the production and degradation rates are kept the same for both models (because production and degradation are energetically costly), but that over all other parameters the precision is optimized. Would that give a qualitatively different result? Why would the gradient range λ be of special importance (the comparison is made on the footing of equal gradient range)? Should not only the precision matter (such that it would be fine that the gradients in the DT and SDC model are different)? In other words, how do the maximal precisions that can be reached given reasonable constraints (such as production and degradation rates) compare, as a function of the distance to the source?

We have significantly altered our description of how we compare the two models to better clarify our process here. Specifically, we now discuss in much greater detail which parameters are equated between the two models and why. The reviewers are correct to point out that the production and degradation rates (*β* and *ν*) should be equated due to their energetic restrictions. While we now present this as our rationale for equating *β*, we also discuss why the explicit value of *ν* is not relevant so long as<inline-graphic mime-subtype="png" mimetype="image" xlink:href="media/image1.png" /> holds true. Thus, while equating *β* and *ν* between models is consistent with our procedure, it still leaves the profile lengthscale *λ*^ˆ^ as a free parameter. Because the profile lengthscale is a frequently measured parameter across a variety of developmental systems, and our goal here is not only to understand the physical mechanisms behind developmental precision but also to quantitatively compare to data, we do not optimize over *λ*^ˆ^. Instead we imagine that the profile lengthscale is set by a functional objective of the developing system, such as where a boundary is to form, and we ask, given this constraint, which model has the higher precision.

7) And in making this comparison, does the SDC model exhibit an optimal diffusion constant that maximizes the precision? Understanding this is probably important, because when, in the current comparison, the SDC model is more precise, it is because of the higher protein clearance rate due to diffusion. Yet, a higher diffusion constant also lowers the steepness of the gradient, decreasing the precision,

Indeed, there is an optimal value of the diffusion constant (and thus *λ*^ˆ^) in the SDC model, for the precise reasons outlined by the referee. However, this optimal value depends on the choice of cell position: a smaller diffusion coefficient maximizes precision at nearby cells, whereas a larger diffusion coefficient maximizes precision at faraway cells. For the same reasons as outlined in the previous response, we do not presuppose the position at which the maximal precision is desired. Instead, we vary *λ*^ˆ^, which allows us to compare the two models systematically and then compare the results to the experimental data.

8) The key mechanism by which the SDC model can become superior is indeed diffusion. Could adding diffusion to the DT model improve the performance of the latter model? Is there evidence for diffusion in experimental systems that employ directed transport?

We are unaware of any experimental study into a possible hybrid system that incorporates both diffusion and directed transport. Nonetheless, we have added a brief comment on such a possibility in our Discussion section.

9) In DT systems, is it clear that molecule transport is independent? How could correlated transport be included in the DT model?

To our knowledge, there have not been detailed experimental studies into the correlation between dynamics of individual morphogen molecules within cytonemes. Introducing correlated transport would violate a key assumption of our argument presented in our new Figure 2 and would create nonlinear and most likely unsolvable noise terms. As such, we consider the investigation into correlated transport to be beyond the scope of this particular work.

10) Also, in the DT model could noise be reduced by cells measuring arrivals of particles rather than averaging over internal concentration of particles? This would presumably only change the resulting expression for precision by a numerical factor, but could this factor shift the balance in favor of DT in some cases?

Measuring the arrival rate of molecules in the DT model would at best reduce the variance by a factor of 2, as the noise from degradation would be eliminated but the noise from arrival would remain. While this would increase the precision of the DT model, it would not do so significantly enough, and our results remain almost entirely unchanged by its inclusion. We have added a footnote in the manuscript to address this possibility.